# Dislocation avalanches are like earthquakes on the micron scale

Péter Dusán Ispánovity [1✉], Dávid Ugi [1✉], Gábor Péterffy [1], Michal Knapek[2], Szilvia Kalácska[1,3], Dániel Tüzes[1], Zoltán Dankházi [1], Kristián Máthis [2], František Chmelík [2] & István Groma[1]

Compression experiments on micron-scale specimens and acoustic emission (AE) measurements on bulk samples revealed that the dislocation motion resembles a stick-slip process – a series of unpredictable local strain bursts with a scale-free size distribution. Here we present a unique experimental set-up, which detects weak AE waves of dislocation slip during the compression of Zn micropillars. Profound correlation is observed between the energies of deformation events and the emitted AE signals that, as we conclude, are induced by the collective dissipative motion of dislocations. The AE data also reveal a two-level structure of plastic events, which otherwise appear as a single stress drop. Hence, our experiments and simulations unravel the missing relationship between the properties of acoustic signals and the corresponding local deformation events. We further show by statistical analyses that despite fundamental differences in deformation mechanism and involved length- and time-scales, dislocation avalanches and earthquakes are essentially alike.

[1] Eötvös Loránd University, Department of Materials Physics, Pázmány Péter sétany 1/a., 1117 Budapest, Hungary. [2] Charles University, Faculty of Mathematics and Physics, Department of Physics of Materials, Ke Karlovu 5, 121 16 Prague 2, Czech Republic. [3] Mines Saint-Etienne, Univ Lyon, CNRS, UMR 5307 LGF, Centre SMS, 158 cours Fauriel 42023, Saint-Étienne, France. ✉email: peter.ispanovity@ttk.elte.hu; david.ugi@ttk.elte.hu

It was not until 1934 that the basic mechanism of irreversible (or plastic) deformation of metals was finally understood when Orowan, Taylor and Polányi independently postulated the existence of a specific lattice defect[1–3]. These line-like defects, called dislocations, can move within the crystal lattice leading to the rearrangement of the atoms and, as a consequence, to the plastic shear deformation of the crystal.

Due to the huge dislocation content in macroscopic metallic samples, their deformation usually appears as a smooth process both in space and time. On microscopic scales, however, the picture changes dramatically. Recent micromechanical experiments demonstrated that when the sample diameter is below a couple of μm (depending on the material), deformation becomes strongly heterogeneous. As pioneering compression tests on Ni single crystal micropillars prepared using focused ion beam (FIB) milling revealed, deformation is a sequence of sudden unpredictable strain bursts that are localized to specific crystallographic planes of the sample[4,5]. During these intermittent bursts, dislocations locally disentangle and move quickly for a short period and then form new metastable sub-structures at the end of an event. The burst sizes follow a scale-free distribution that suggests an underlying self-organization of the dislocation structure upon these plastic events[6,7].

A unique experimental method that is able to monitor this stochastic response is the detection of AE waves. The principle of the emission of acoustic waves in materials is analogous to earthquakes: Plastic deformation is caused by the local rearrangement of dislocation lines in a crystal, a process that is strongly dissipative and part of the released elastic energy escapes in the form of elastic waves, that can be detected at the surface[8]. It was found that in bulk ice single crystals the recorded AE signal is burst-like and the energy associated with individual bursts follows a scale-free distribution[9,10]. The found power-law exponent is robust, typically not affected by deformation mode, and for single crystals with hexagonal closed-packed (HCP) structure it was measured to be $\tau_E = 1.5 \pm 0.1$[11]. These kinds of measurements so far have only been performed on bulk samples and it is believed—but not yet demonstrated—that the AE waves are emitted from similar local strain events that can be directly observed only for micron-scale objects.

In this work, we accomplish the task of detecting weak AE waves which arise during micropillar deformation. The main advantage of this approach is that, in this case, AE sources are highly localized within a small micropillar volume that prevents uneven attenuation of AE waves arising in different parts of the specimen, this being inherent to bulk materials testing. Hence, plenty of innate AE waves can be detected due to dislocation slip, which can, in turn, provide interesting and nontrivial insights into the dynamics of plastic events.

## Results and discussion

In order to be capable of recording AE signals originating in the plastically loaded micropillars, the experimental set-up sketched in Fig. 1a was developed (see Supplementary Fig. 1 for a photo). The device can be placed inside a scanning electron microscope (SEM) that allows us to collect three different types of information simultaneously during compression of the micropillars: (i) stress and strain using a capacitive displacement sensor measuring the elongation of a spring, (ii) acoustic signal from a piezoelectric transducer and (iii) visual images using the electron beam of the SEM. The major difficulties for AE detection in micropillars comprised the relatively low number of dislocations involved in the slip process (compared to bulk materials testing) and various sources of noise signals in the SEM chamber, mostly of electromagnetic origin. For further details on the experimental set-up and remedies to these issues see Methods.

Firstly, rectangular micropillars with a 3:1:1 aspect ratio and side lengths of $d = 8 - 32\,\mu m$ were prepared from a Zn single crystal oriented for single slip (for more details on the sample see Methods). In Fig. 1b a micropillar during the course of the experiment is shown. One can observe that dislocation slip indeed takes place solely on the basal plane of the HCP lattice (see also Supplementary Movie 1). Since the crystal orientation in the pillar remains the same throughout the entire loading (see Supplementary Fig. 2) only dislocation glide is operative and deformation due to twinning can be excluded.

Figure 1c plots the measured compressive stress $\sigma$ as a function of time $t$ for the micropillar with $d = 32\,\mu m$ shown in Fig. 1b (see also Supplementary Movie 1 for an in situ video of the compression). The pronounced close-to-vertical drops correspond to the strain bursts that lead to the sudden elongation of the spring of the device.

To analyse the spatial distribution of a strain burst two consecutive SEM images taken before and after the stress drop highlighted with grey color in Fig. 1c were compared with edge detection on the differential image and we concluded that deformation took place solely in a thin slip band highlighted with red in Fig. 1b. During the compression, AE signal is also recorded that comprises numerous individual bursts and their rate exhibits robust correlation with the stress drops (Fig. 1c). To elaborate further on this finding Fig. 1d, e plot consecutively zoomed parts of the stress-time curve shaded with grey colour. According to Fig. 1d AE events can only be detected when plasticity occurs, that is, when the stress-time curve deviates from the linear ramp-up characteristic of purely elastic deformation. Interestingly, it is possible that several AE events correspond to the same stress drop as also indicated by the event count number (Fig. 1d). The reason for this is that the data acquisition rate differs considerably between stress (200 Hz) and AE (2.5 MHz) measurements, the latter allowing for a more detailed analysis. Figure 1e shows that the AE signal consists of short ($\lesssim 100\,\mu s$) peaks standing out from the background noise. We, thus, conclude that the AE events are indeed due to the dislocation activity leading to plastic slip within the micropillar, however, the abundance of AE events suggests that a measured stress drop is a result of complex internal dynamics on timescales not accessible by stress measurements.

**Origin of AE events**. To quantify the correlation between plastic deformation and AE we now turn to the statistical analyses of the measured data. In agreement with studies on other single crystalline micropillars the distribution of the size of the individual stress drops $\Delta\sigma$ follows a scale-free distribution with a cut-off $\sigma_0$: $P(\Delta\sigma) \propto \Delta\sigma^{-\tau_\sigma} \exp(-\Delta\sigma/\sigma_0)$ (Fig. 2a)[6,7]. According to the inset, if the axes are re-scaled with the cross section $A = d^2$ of the micropillars (that is, force drop $\Delta F = A\Delta\sigma$ is considered as variable) the curves overlap and can be fitted with a master function yielding $\tau_\sigma = 1.8 \pm 0.1$ and $F_0 = 1.5 \pm 0.1$ mN for the exponent and the cutoff, respectively. Note that noise of the force measurement prohibits the reliable detection of drops below ~0.1 mN. The distribution of the AE event energy $E$ is characterized by another scale-free distribution now without an apparent cut-off and dependence on pillar size: $P(E) \propto E^{-\tau_E}$ (Fig. 2b) with $\tau_E = 1.7 \pm 0.1$. Note that the recorded AE events were, in general, well-defined in time, with no significant effect of signal overlapping or reflections (see Methods), as often observed in bulk samples.

The facts that (i) stress drops $\Delta\sigma$ and AE energies $E$ are detected in a correlated manner, (ii) both obey a scale-free distribution and (iii) the exponents are relatively close to each other suggest that there is a physical relation between them. To shed light on such a link, Fig. 2c provides a scatter plot of the *injected energy* $E_{inj}$ and the detected AE energy $E$ corresponding

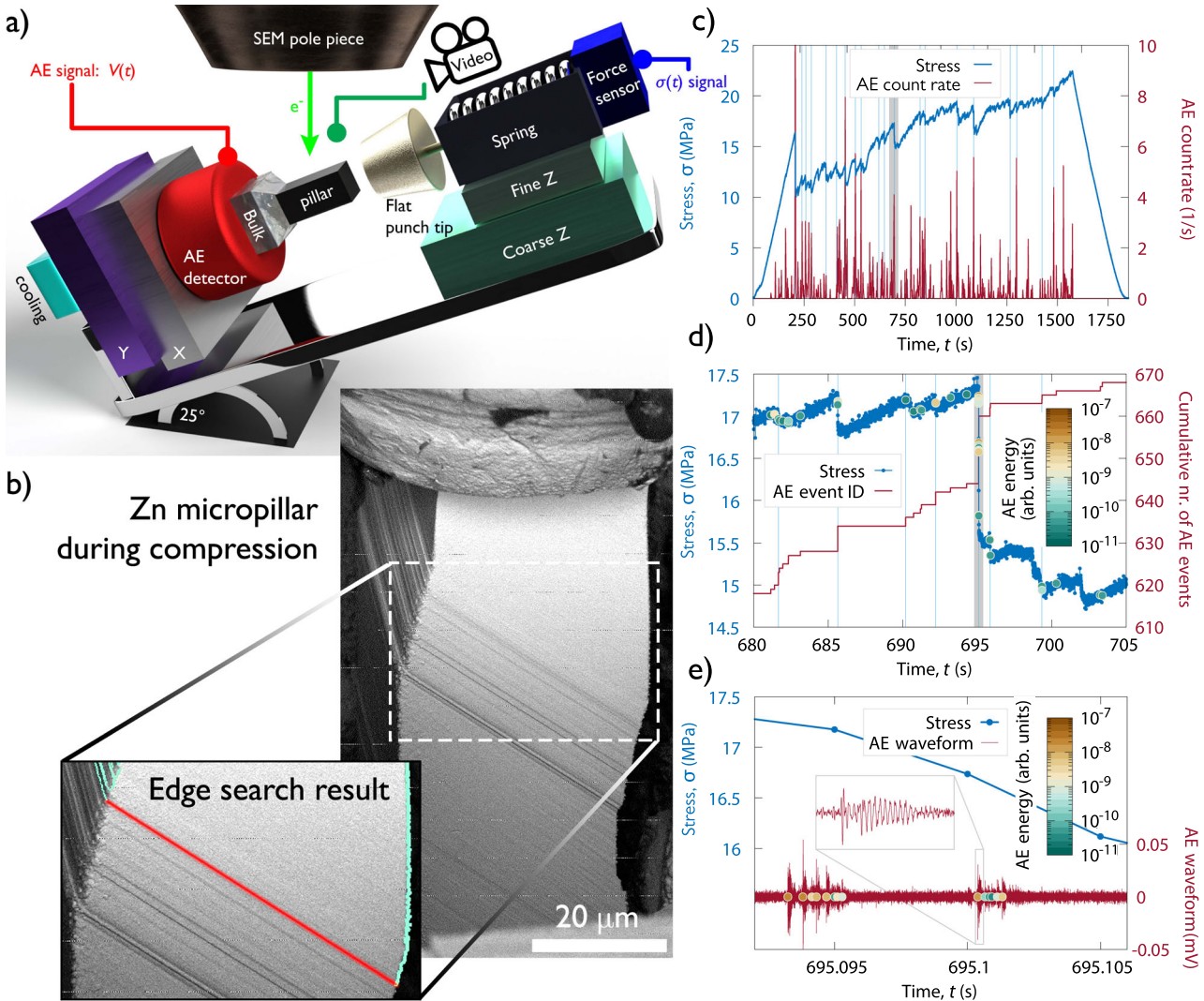

**Fig. 1 Compression experiment of Zn micropillars oriented for single slip. a** Sketch of the experimental set-up with a disproportionately large micropillar for clarity. **b** Backscattered electron image of a $d = 32\,\mu m$ micropillar during compression. The magnified image shows the slip band in red corresponding to the stress drop highlighted in grey in panels (c) and (d). The location of the band was obtained by edge search on SEM images before and after the stress drop. **c** Measured stress vs. time as well as the averaged rate (obtained by convolution with a Gaussian of 0.5 s width) of the detected individual AE bursts. The light blue vertical lines mark the stress drops larger than 1 MPa. **d** Zoomed stress-time curve of the region shaded by grey in panel (c). The coloured data points along the stress curve represent the individual AE events and their energies, whereas the red curve shows the cumulative number of these events. The light blue vertical lines mark short periods with at least two AE events. **e** Zoomed stress-time curve of the region shaded in grey in panel (d) and the detected AE waveform of the same interval. The inset shows the magnified view of a single event and coloured data points correspond to individual signals detected by thresholding the AE signal.

to the individual stress drops (given that at least one AE event was detected during the stress drop) for $d = 32\,\mu m$ pillars (for smaller pillar sizes see Supplementary Fig. 3). The injected energy refers to the work done by the compression device during a stress drop and is proportional with $E_{inj} \propto (\sigma - \Delta\sigma/2)\Delta\sigma$ with $\sigma$ being the applied stress at the onset of the event (see Energetic considerations in Methods for details and background discussion). As said above, several AE events may be detected during a single drop. In such cases the energies of the corresponding AE events are added. As seen, there is a large scatter between $\Delta\sigma$ and $E$ but, clearly, stress drops with larger injected energy $E_{inj}$ tend to emit AE signals with larger energies also expressed by the Pearson correlation found to be $0.5 \pm 0.1$. If one, however, bins the data with respect to the injected energy a clear close-to-linear dependence $E \propto E_{inj}$ is obtained between the two quantities. This means that although a one-to-one correspondence between $E$ and $E_{inj}$ does not exist, there is a linear relationship in the average

sense that allows one to obtain the distribution of the local energy release (that is proportional to the plastic strain increment multiplied with the local stress during a local deformation event) from the statistics of the AE events.

**Aftershock and foreshock statistics.** As mentioned above, AE signals emitted by local plastic deformation events are similar to elastic waves caused by the seismic activity in the Earth's crust (although they differ in their amplitude and frequency spectra by several orders of magnitude). To deepen the analogy, we now continue with the analysis of AE signals that offer a much better time resolution than the stress measurements. We intend to assess whether the AE bursts obey the three ubiquitous fundamental scaling laws associated with earthquakes. (i) The Gutenberg-Richter law[12] states that the probability density of an earthquake with released energy $E$ decays as a power-law[13]:

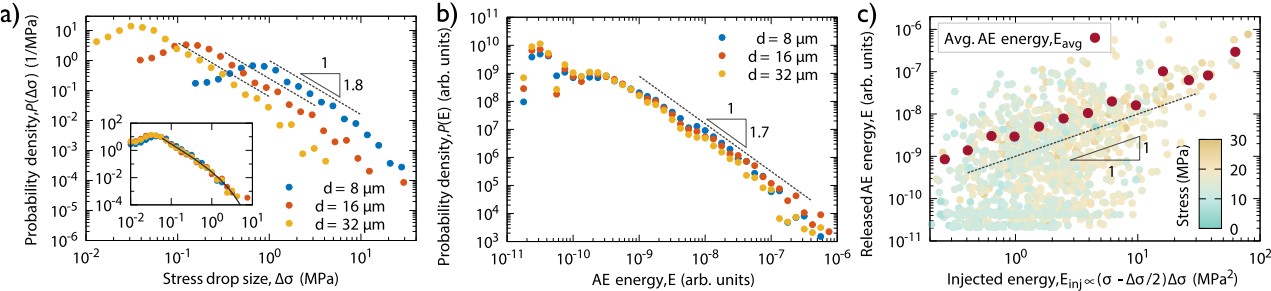

**Fig. 2 Correlation between the stress drops and the acoustic signals. a** Distribution of stress drop sizes $\Delta\sigma$ for different pillar side lengths $d$. The probability density functions (PDFs) follow a power-law with exponent $\tau_\sigma = 1.8 \pm 0.1$. The inset shows the PDF as a function of the force drop $\Delta F = \Delta\sigma \cdot d^2$ with units in mN. The collapsed curves can be fitted with a master function above the detection threshold and exhibit a cut-off at $F_0 = 1.5 \pm 0.1$ mN. **b** Distribution of AE energies of individual signals detected at the sample surface. The curves are characterized by a power-law exponent $\tau_E = 1.7 \pm 0.1$ and do not exhibit an apparent cut-off and do not depend on the pillar side length $d$. **c** Scatter plot of the injected energies $E_{\text{inj}}$ during stress drops of $d = 32\,\mu\text{m}$ pillars and the corresponding summed released AE energies $E$. The color-scale refers to the actual stress at which the stress drop took place along the stress-time curve and do not show correlation with the injected energy. The red dots represent the average released energies $E_{\text{avg}}$ obtained by averaging the datapoints for bins of logarithmically increasing width. The dashed line represents the $E \propto E_{\text{inj}}$ linear relationship.

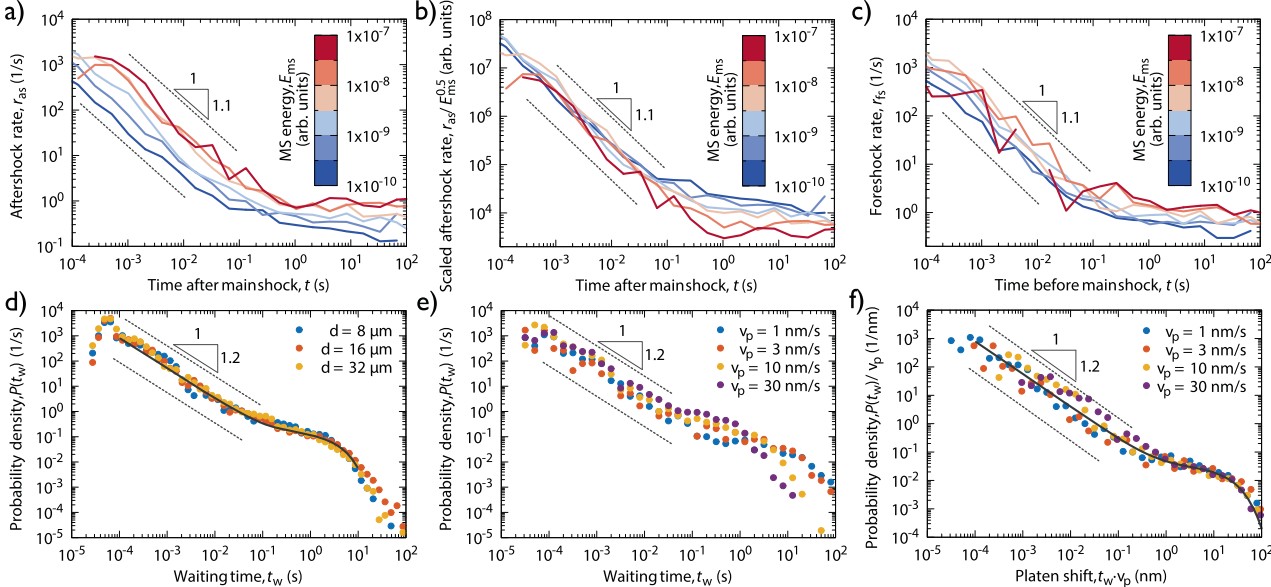

**Fig. 3 Temporal statistical analyses of AE events. a** The rate of aftershocks $r_{\text{as}}$ after a main shock with an energy given by the colour for $d = 32\,\mu\text{m}$ pillars (Omori law). **b** Curves of panel a) divided with the square root of the main shock energy $E_{\text{ms}}$ (aftershock productivity law). **c** Rate of foreshocks $r_{\text{fs}}$ before a main shock of energy given by the colours for $d = 32\,\mu\text{m}$ pillars (inverse Omori law). **d** PDF $P(t_w)$ of waiting times $t_w$ between subsequent AE events for pillars of various sizes. **e** $P(t_w)$ for $d = 8\,\mu\text{m}$ pillars and different platen speeds $v_p$. **f** $P(t_w)$ re-scaled with the platen velocity $v_p$. Note that the minimum $t_w$ of 20 μs, i.e., the minimum time between two subsequent AE events, is defined as one of the AE event individualization parameters (see Methods).

$P(E) \propto E^{-w}$ with $w \approx 5/3$. (ii) According to the Omori law, the rate of aftershocks $r_{\text{as}}$ after a main shock decays approximately inversely with the time $t$ elapsed[14,15]: $r_{\text{as}}(t) \propto t^{-p}$ with $p \approx 1$. (iii) The aftershock productivity law in seismology concludes that main shocks with larger energy $E_{\text{ms}}$ produce on average more aftershocks: $r_{\text{as}} \propto E_{\text{ms}}^{2\alpha/3}$ with $\alpha \approx 0.8$ found empirically[16]. The existence of scale invariance through these power-law relationships has also been demonstrated in laboratory-scale compression experiments on porous bulk materials[17] and rocks[18].

It is found by our analysis that, despite the huge difference in spatial and temporal scales, the deformation mechanisms and the mode of loading, all three scaling laws are found to hold for micropillars, too. The Gutenberg-Richter law was demonstrated in Figs. 2b and 3a proves the Omori law for $d = 32\,\mu\text{m}$ pillars. The line colours refer to the energy of the main shock, and it is clear that the rate indeed decays as a power-law with $p = 1.1 \pm 0.1$ for approx. three decades and then saturates likely

due to the onset of new sequences. In accordance with the productivity law the rate is larger for larger main shocks and collapse can be obtained by re-scaling the rate with $E_{\text{ms}}^{0.5}$ (Fig. 3b), yielding $\alpha = 0.75$. As a further proof of equivalence, Fig. 3c plots the correspondent of the inverse Omori law describing the power-law increase in the rate of foreshocks $r_{\text{fs}}$ before a main shock[19]. (Supplementary Fig. 4 shows the corresponding figures for smaller pillars.) Previously, a similar analysis of AE on hexagonal ice at the bulk scale also showed more abundant triggering after large events, but the scale-free characteristics presented here were not possible to obtain, likely because of the relatively high level of noise[20]. It is also noted, that the difference between the foreshock and aftershock rates is significantly smaller compared to that of earthquakes. It is speculated that the difference is caused by the ambiguity in the determination of the main shock due to the noise present in the AE energy measurement.

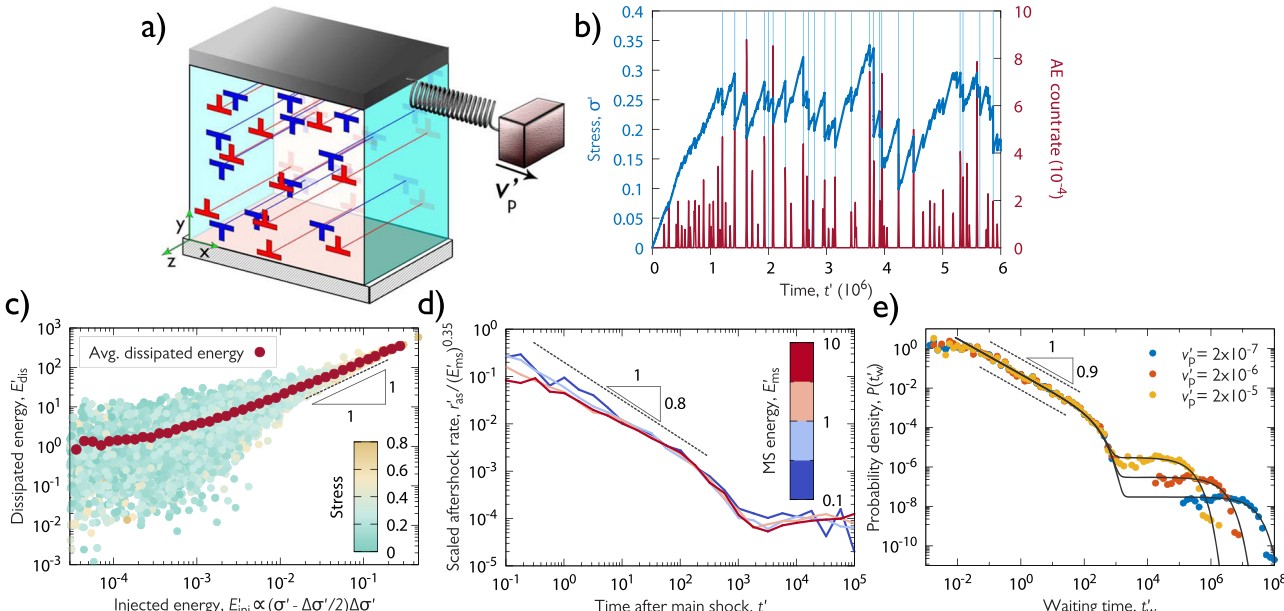

**Fig. 4 DDD simulations. a** Sketch of the simulation set-up. The system is infinite in direction $z$ and periodic boundary conditions are applied in directions $x$ and $y$. **b** Stress vs. time curve as well as the averaged rate of the simulated individual AE bursts for a representative configuration. The light blue vertical lines show the stress drops larger than 0.02. **c** Scatter plot of the injected energies during stress drops and the corresponding summed released AE energies for systems of $N = 1024$ dislocations, see caption of Fig. 2c for details. **d** The rate of aftershocks $r'_{as}$ scaled with $(E'_{ms})^{0.35}$ after a main shock with energy $E'_{ms}$ given by the colour for $N = 1024$ dislocations (Omori and productivity laws). **e** PDF $P(t'_w)$ for $N = 256$ dislocations and different platen speeds $v'_p$.

The distribution of waiting times $t_w$ between subsequent AE events, a key measure of temporal correlations and clustering in temporal processes[21,22], was also analysed. For earthquakes a universal gamma distribution upon re-scaling with the seismic occurrence rate was reported[23]. A similar distribution is found here (Fig. 3d): $P(t_w) = [At_w^{-(1-\gamma)} + B]\exp(-t_w/t_0)$, which can be interpreted as follows. The power-law decay for small ($\lesssim 0.1$ s) waiting times corresponds to the correlated temporal clusters originating from the same plastic event, often observed as a single stress drop. The exponent $1 - \gamma = 1.2 \pm 0.1$ coincides with the Omori exponent $p$ within error margins, as expected. For larger times a plateau with an exponential cut-off is observed corresponding to a Poisson-like process of uncorrelated signals coming from different plastic events. To confirm this hypothesis we repeated the experiments for the $d = 8\,\mu$m pillars with different platen velocities $v_p$ (i.e., deformation rates). Whereas the single event dynamics (power-law part) is unaffected by the velocity $v_p$ (Fig. 3e), the collapse of the curves in the cut-off region after re-scaling the axes with the velocity $v_p$ (Fig. 3f) yields $t_0 \propto v_p^{-1}$ and $B \propto v_p$.

These results unveil an interesting two-level structure of plastic activity: accumulation of plastic strain is characterized by the intermittent appearance of uncorrelated slip bands. These strain bursts induce stress drops due to the stiffness of the compression device. But AE measurements reveal that these strain bursts themselves are characterized by a sequence of local events with complex spatiotemporal dynamics. Firstly, Omori law and the waiting time distributions report about scale-free temporal correlations. Secondly, since strain increments during stress drops are localized in distinct slip bands (see Supplementary Movie 1) and plastic activity was found to be responsible for AE, one can conclude that the correlated AE events originate from a single slip band, that is, they are not only temporally but also spatially correlated. To quantify this observation the strain evolution between subsequent SEM images during the in situ compression was analyzed in the Methods (see section Strain

localization) and we concluded that deformation during a single stress drop is indeed highly localized and is typically concentrated in one or sometimes few individual slip bands.

As seen in Fig. 2a the stress drop size distribution exhibits a cut-off, that is, correlated consecutive triggering during a strain burst is limited. The behaviour of the cut-off, thus, may shed light on the physics of the triggering mechanisms. As it was mentioned, the cut-off in stress $\sigma_0$ decreases with increasing pillar side length $d$, but the cut-off in force $F_0 = d^2\sigma_0$ is independent of $d$. According to Csikor et al. the physical origin of the cut-off is either elastic coupling with the compression device (during an event the applied stress drops that reduces the driving force of the event) or the strain hardening of the material (during an event plastic deformation makes the material harder so the driving force drops in a relative sense)[7]. From the comparison of the behaviour of the cut-off with the predictions of Csikor et al. we conclude that it is not the machine stiffness but rather the local strain hardening in the activated slip band is responsible for stopping the consecutive triggering taking place during a stress drop (see Methods for a detailed discussion).

Recently, Houdoux et al. investigated the plastic response of granular systems that also showed remarkable analogy with earthquakes[24]. They concluded that in the triggering of subsequent events it is not the time but rather the strain that matters. In our case, however, because the exponent $1 - \gamma$ is rather close to one, one cannot decide whether time (Fig. 3e) or strain (Fig. 3f) matters in the triggering mechanism.

**Numerical modelling.** To provide a possible physical explanation for the experimentally observed behaviour we conduct discrete dislocation dynamics (DDD) simulations of parallel straight edge dislocations gliding on a single glide plane (see sketch in Fig. 4a). Deformation of Zn micropillars is predominantly single slip, yet, the computational model is a simplification of the realistic system as it neglects, e.g., curvature and applies different boundary conditions (see Methods for details). However, it captures

properly the long-range stress field of dislocations that was shown to play an essential role in the critical behaviour of dislocations[9,25]. Since no length-scale other than the average dislocation spacing and the system size is present (due to the scale-free $1/r$-type dislocation interactions) dimensionless variables denoted with $(\cdot)'$ are introduced hereafter (see Methods and Supplementary Table 1)[26,27].

A loading method analogous to the micropillar experiments is implemented, i.e., a platen is moved with velocity $v'_p$ and the load is transferred to the system via a spring. As a result, dislocation avalanches appear as stress drops here as well (Fig. 4b). During the avalanches, dislocations move rapidly, and due to the overdamped dynamics assumed for dislocations ($v' \propto F'$, where $v'$ and $F'$ is the velocity and the acting force for dislocations, respectively) the elastic energy release rate reads as $\sum_i v_i'^2$, with the sum performed over all dislocations. By thresholding this rate one can emulate the sensitivity of the AE sensor and obtain simulated AE events as well as the corresponding released energies (see Methods). Like in the experiments, the simulated AE events show strong correlation with the stress drops (Fig. 4b, Supplementary Movies 2, 3). It has been known that size distribution of dislocation avalanches exhibits a different exponent in simulations compared to the real samples[25,28], yet, the temporal clustering of the simulated AE events shows very similar behaviour to experiments in terms of the correlation between injected energies and the AE energies (Fig. 4c), Omori law (Fig. 4d), and waiting time distribution (Fig. 4e). We thus conclude, that the complex dynamic behaviour observed in the experiments reported in this paper is the result of the spatio-temporal correlations of the dislocations due to their long-range elastic interactions and the lack of short-range mechanisms, such as dislocation reactions.

It has always been the fundamental assumption of AE experiments that the parameters of the signals are characteristic of the local deformation process. The experiments and simulations reported here prove this long-standing hypothesis and reinforce that intermittency and scale-invariance characterizing plastic deformation of HCP single crystals are related to the self-organized critical (SOC) behaviour of dislocations. In addition, we showed that plastic events, similarly to earthquakes, do not only exhibit spatial but also temporal clustering with long-range correlations, however, the differences in involved length and timescales can be as large as nine orders of magnitude, as summarized in Supplementary Table 2. This phenomenon also raises analogy with many other physical systems exhibiting crackling noise[29]. It is known, however, that SOC behaviour is not ubiquitous in crystal plasticity, for instance, it is suppressed in materials with FCC and BCC crystal structure and under multiple slip conditions likely because of short-range interactions related to dislocation reactions and also at high temperatures[30–32]. Dedicated further experiments and modelling based on the new methodologies of this paper are needed to study and understand whether dislocation dynamics is altered under such circumstances in terms of magnitude and spatiotemporal distribution of plastic fluctuations.

## Methods

**Sample preparation**. High purity single crystalline zinc heat treated at 100 °C for 4 h under atmospheric air, oriented for basal slip with side orientation corresponding to the $\langle 2\bar{1}\bar{1}0 \rangle$-type normal direction (Supplementary Fig. 2) was mechanically polished sequentially with SiC grinding paper and alumina suspension (down to 1 μm). This was followed by a fast (10 s) electropolishing with Struers D2 solution at 20 V, 1 A. A sharp perpendicular edge was then created on the bulk specimen by low energy Ar ion polishing (5 kV, 2 mA).

Experimental work including micropillar milling, EBSD measurements and micromechanical testing was carried out inside the vacuum chamber of an FEI Quanta 3D dual beam scanning electron microscope (SEM). Focused ion beam

(FIB) operating with Ga⁺ ions was used to fabricate square-based pillars of various sizes (8 μm: 13 pieces, 16 μm: five pieces and 32 μm: four pieces with an approximate 3:1 aspect ratio of height to side), with final beam conditions of 30 kV, 1–3 nA. In order to minimize Ga⁺ ion contamination on the surface and create practically non-tapered (≤2.5° between the side and the loading axis) samples, the pillars were fabricated in a lathe milling configuration[33]. On the top of the pillars a thin (~350 nm) Pt cap was deposited by FIB to act as hard buffer material between the pillars and the flat punch tip and also to reduce ion contamination during FIB-milling.

### Analytical methods

*Microstructure analysis*. For electron backscatter diffraction (EBSD) measurements, the Edax Hikari camera was used with 1 × 1 binning, and the OIM Analysis v7 software provided the orientation results. Unit cell corresponding to the cross-sectional side of the pillar can be seen in Supplementary Fig. 2. To calculate the initial geometrically necessary dislocation (GND) density, a digital image cross-correlation based technique called high (angular) resolution electron backscatter diffraction (HR-EBSD) was applied[34]. HR-EBSD determines local strain and stress tensor components with the help of the raw diffraction patterns (for details see ref. [34]). This method requires a reference diffraction pattern for the image correlation, that is ideally captured in the strain-free state of the lattice. A perfect reference pattern is often difficult to obtain experimentally, therefore in our case a pattern with the presumably lowest stress is chosen, creating a relative scale for the GND density. Diffraction patterns were recorded with approx. 500 × 500 px² resolution from an area of 16.2 × 14.2 μm² with a step size of 100 nm. The evaluation was carried out by BLGVantage CrossCourt v4.5 software. 20 regions of interest (of 128 × 128 px² each, Supplementary Fig. 5) were selected from each diffraction pattern to carry out the HR processing with applied high and low pass filtering. All points in the map were evaluated. The estimated average value of $\rho^{GND} = 1.2 \times 10^{13}$ m⁻² was measured on a surface prepared by the same FIB conditions (30 kV, 3 nA) as it was used for the pillar fabrication prior to deformation. This value is close to the detection limit of the GND density by HR-EBSD, hence it is concluded that the sample preparation did not introduce a significant/measurable dislocation content in the sample.

*X-ray line profile analysis*. Dislocation density characterization by X-ray diffraction measurements was performed on the bulk Zn single crystal sample prior to the micropillar fabrication. The X-ray line profiles of the $(10\bar{1}1)$ reflection were obtained by a double-crystal diffractometer using Cu Kα radiation (Supplementary Fig. 6a). The experimental setup is of θ −2θ type, that consists of a high intensity Rigaku RU-H3R rotating anode X-ray generator with a copper anode, a mono-chromator that filters out the Cu Kα₂ component and redirects the X-ray beam to the sample, and the Dectris MYTHEN 1D wide range solid state X-ray detector that records the peak at a distance of 960 mm. We also used a cylindrical vacuum chamber between the sample and the detector in order to increase the peak-to-background ratio. The quantification of the total dislocation density was carried out by the variance method[35,36] by analyzing peak broadening based on the asymptotic behaviour of the second order restricted moment:

$$M_2(q) = \frac{1}{\pi^2 \epsilon_F} q + \frac{\Lambda}{2\pi^2} \langle \rho \rangle \ln \frac{q}{q_0}, \tag{1}$$

where $q = 2(\sin\theta - \sin\theta_0)/\lambda$, $\lambda$ corresponds to the wave length of the applied X-rays, and $\theta$ and $\theta_0$ are half of the diffraction and Bragg angles, respectively. Parameter $q$ corresponds to the distance from the peak center in reciprocal space, $q_0$ is a constant depending on the dislocation-dislocation correlations, $\epsilon_F$ is the coherent domain size, and $\langle \rho \rangle$ is the average dislocation density. The value of $\Lambda$ is commonly given as $\Lambda = \pi|\mathbf{g}|^2|\mathbf{b}|^2 C_\mathbf{g}/2$, where $\mathbf{b}$ and $\mathbf{g}$ are the Burgers and diffraction vectors, respectively, and $C_\mathbf{g}$ is the diffraction contrast factor that depends on the type of dislocations in the system and on the relative geometrical position between the dislocation line direction $\mathbf{l}$ and the direction of $\mathbf{g}$[36,37].

For this reason for the determination of the initial dislocation density one has to make assumptions about the relative densities of dislocations of different types. Since the energy of dislocations with Burgers vector lying in the basal plane is lower compared to other types it is reasonable to assume that in the original undeformed sample each slip system with a Burgers vector lying in the basal plane is equally populated. To account for elastic anisotropy the corresponding average $\Lambda$ was determined numerically by the ANIZC program (http://metal.elte.hu/anizc/program-hexagonal.html) using the elastic moduli of Zn, yielding $\Lambda = 0.506$[38,39].

As the coherent domain size is larger than ~1 μm, the first term in Eq. (1) is negligible. As a result of the second term caused by the dislocations, $M_2$ versus ln($q$) plot indeed becomes a straight line in the $q \to \infty$ asymptotic regime, as shown in Supplementary Fig. 6b. From the fit a total dislocation density of $\langle \rho \rangle^{XRD} = (1.5 \pm 0.1) \times 10^{14}$ m⁻² was obtained. As expected, this value is higher than the GND density determined by the HR-EBSD technique, therefore it can be assumed that the initial dislocation network mostly consisted of statistically stored dislocations.

### Micromechanical experiments

*Testing device*. Room temperature compression tests on the micropillars were carried out in high vacuum mode inside the SEM chamber to allow in situ

monitoring of the deformation process and slip activity on the pillars' surface by secondary and backscattered electrons. A custom-made nanoindenter[40,41] shown in Supplementary Fig. 1 was used without any load or strain feedback loop integrated. The precision of the indentation depth and load was ~1 nm and ~1 μN, respectively. The applied sampling rate was 200 Hz, while platen velocity (if not stated otherwise) and spring constant were 10 nm/s and 10 mN/μm, respectively. For a detailed description of the device, the reader is referred to ref. [40]. Exemplary stress-strain curves are presented in Supplementary Fig. 7. The curves show the intermittent nature of plasticity in micropillars and also provide evidence of the so-called plasticity size effect ('smaller is harder').

*Cut-off analysis*. In this section background discussion is provided on the physics of the subsequent triggering taking place during an event cluster that is seen as a single stress drop. During stress drops the driving force (that is, the applied stress itself) gets smaller that is expected to reduce the probability of subsequent triggering. In particular, the stress drop cannot be larger than the actual stress itself, so, there is definitely a hard barrier related to the stress. However, according to Fig. 2a the stress drops get smaller for larger systems that indicate that the stress may not be the limiting factor for avalanche propagation. As shown by the inset, the cut-off in force $F_0$ is independent on the specimen height $L = 3d$, which then also holds for the elongation decrements $x_0 = F_0/k$, with $k$ being the spring constant of the device. The effect of machine stiffness on the avalanche cut-off was investigated by Csikor et al.[7], they found that the cut-off in strain obeys scaling:

$$\varepsilon_0 \propto \frac{bE}{L(\Theta + \Gamma)}, \qquad (2)$$

where $\Theta$ and $\Gamma$ are the strain hardening coefficient and the machine stiffness, respectively. In our case the machine stiffness is $\Gamma \approx k/L$, since the elastic deformation of the pillar is negligible compared to that of the spring. Hence, the cut-off in force reads as

$$F_0 = kx_0 = kL\varepsilon_0 \propto bE \frac{k}{\Theta + k/L}. \qquad (3)$$

The finding that $F_0$ does not depend on $L$ suggests that $\Theta$ is significantly larger than $k/L$. But if we look at the approximate values we see that $k/L \approx 100$ MPa (for a 32 μm pillar) and the average slope of the stress-strain curves is around 50 MPa. To overcome this apparent contradiction we consider the origin of $\Theta$ in the scaling relation above. During a stress drop not only the stress decreases but also the material gets harder (strain hardening), both processes act to cease the event. This is the reason the sum of $\Theta$ and $\Gamma$ appear above in Eq. (2). In ref. [7] $\Theta$ was identified with the slope of the stress–strain curve. We believe, however, that this value is a local quantity and may differ from the global slope. During the compression of a pillar deformation proceeds in different shear bands. This can be envisaged as a weakest link process, that is, always the shear band with the lowest yield stress gets activated. After activation, strain accumulates but the deformation stops and another shear band will get activated subsequently. This means that the yield stress of the activated shear band increases, that is, strain hardening takes place. This hardening coefficient of this local mechanism is nothing to do with the global coefficient, that also depends on the number of shear bands and the distribution of the yield stresses of the shear bands. So, we argue, that the $\Theta$ in Eq. (2) may be significantly larger than the global strain hardening coefficient. This could explain why $F_0$ is independent of the system size and suggests that the local hardening mechanism is dominant in the avalanche cut-off over the stress decrease due to the applied spring. Whether local hardening is due to dislocation accumulation or dislocation starvation through the surface is an open question that future TEM investigations are expected to answer.

*Edge detection*. In order to investigate the spatial distribution of the plastic strain corresponding to individual stress drops, edge detection was performed sequentially on each SEM image of the $d = 32$ μm micropillar shown in Fig. 1b. We aimed at detecting the vertical edge on the right side of the micropillar as it was characterized by a large difference in the intensity in the horizontal direction (due to the dark background). First, a vertical line was selected in the middle of the pillar as a reference. To detect the sudden change in intensity the pictures were then processed row by row starting from the reference line. If the drop in the intensity was larger than the given threshold, the point was marked as part of the edge. The horizontal coordinate $x$ obtained at the height of $z$ is denoted as $x_{raw}(z)$. The raw images were processed using the OpenCV package[42].

The used backscattered electron detector introduced high intensity noise in the form of short horizontal lines with a width of few pixels, which needed to be filtered. Noise filtering was, thus, applied on $x_{raw}(z)$ with moving median smoothing. The window size was selected to be 7 pixels both upwards and downwards and if the current pixel along the $x$ axis deviated for more than 2 μm it was replaced by the median. The filtered curves are denoted as $x(z)$. Supplementary Movie 4 shows how the algorithm works during the course of the experiment.

The time development of $x(z)$ is shown in Supplementary Fig. 8. The base of the sample was moved to the origin and the results were rotated by one degree clockwise. The white gaps represent strain bursts when large plastic deformation occurs between consecutive images. The slip band can be located by determining the end of the gap. As seen, the gaps end at well-defined points, confirming that strain bursts take place within 'thin' slip bands.

Based on Supplementary Fig. 8, the SEM images recorded before and after the stress drop analysed in Fig. 1c–e were identified and the corresponding edge shapes were denoted by purple and pink colours, respectively. These SEM images are shown in Supplementary Fig. 9a, b. Although it is barely seen by visual inspection, the quantified difference of the two images $\Delta x_{raw,t}(z,t) = x_{t+\Delta t}(z) - x_t(z)$ (Supplementary Fig. 9c) proves that deformation took place along the slip plane at the height of ~28 μm (as also seen as horizontal grey line in Supplementary Fig. 8 and highlighted by a red line along the corresponding basal plane in Fig. 1b).

*Strain localization*. In order to quantify how local deformation between two consecutive SEM images was we stared from the $\Delta x_{raw}(z)$ curves obtained in the previous section. These curves were still rather noisy. We, therefore, calculated the moving average with a window size of 15 pixels and then applied a moving median smoothing with a window size of 61 pixels. Finally, we made the curves monotonous, since slip in the opposite direction was not observed in the experiments. Three representative exemplary so obtained $\Delta x(z,t)$ curves can be seen in the bottom row of Supplementary Fig. 10.

The obtained profiles usually exhibit a single slip band, but sometimes more than one step in the profile is seen. To quantify to what extent is the deformation localized we use the method of ref. [43]. We first note, that $\Delta x(z,t)$ is defined on an equidistant grid of the individual pixels of the SEM image. Let the discretized profile be denoted as $\Delta x_i$ (for simplicity we omit the reference to time $t$). The local strain increment is then $\Delta \varepsilon_i^{pl} = \Delta x_{i+1} - \Delta x_i$. We now select an arbitrary point with index $k$ along the height of the pillar and consider the typical distance of the plastic strain increments from this point as:

$$d_k = \frac{\sum_i \Delta \varepsilon_i^{pl} |k - i| \Delta z}{\sum_i \Delta \varepsilon_i^{pl}}, \qquad (4)$$

where $\Delta z$ is the pixel size of the SEM image. Then the minimum $d_{min} = \min_k d_k$ is determined. For a homogeneous distribution of the plastic strain (i.e., $\Delta x(z)$ is a linear function) $d_{min}$ equals $L/4$, where $L$ is the height of the micropillar. On the other hand, for a fully localized strain distribution (i.e., $\Delta x(z)$ is a step function) $d_{min} = 0$ is obtained. The localization parameter is, therefore, defined as

$$\eta = 1 - \frac{4}{L} d_{min}. \qquad (5)$$

Consequently, $\eta = 0$ signals a homogeneous deformation, whereas $\eta = 1$ is characteristic of deformation fully localized in a single slip band.

Supplementary Fig. 10 summarizes the analysis performed on those consecutive images, where the event size defined as $\Delta x(L) - \Delta x(0)$ (that is, the displacement between the top and bottom of the pillar) was larger than 0.02 μm. As seen the localization $\eta$ is typically between 0.5 and 1 and its average is $\langle \eta \rangle = 0.74$. This high value of $\eta$ clearly shows that plastic strain increments are quite localized. The fact that the values are smaller than 1 are likely due to the numerical noise present in the edge detection and that between two consecutive SEM images (that takes around 0.25 s) more than one events can take place.

So, based on this analysis we conclude that the plastic strain increments are highly localized, typically concentrating in a single slip band. Since the AE events are observed during the accumulation of plastic strain, it is natural to assume that a cascade of events correlated in time originate from the same (typically one, sometimes few) slip band. This means these events are not only correlated in time, but also in space. More details on the spatial correlations are not possible to obtain with the present experimental methods due to the small volume of the specimen.

## AE measurements

*Detecting AE signals*. By definition, acoustic emissions are transient elastic waves generated in materials due to sudden localized and irreversible structure changes[44,45]. The detection of AE waves is based on its physical nature – when the material is subjected to external loading, released energy forms stress pulses propagating through the material as transient elastic waves. The wave component perpendicular to the surface is detected typically by a piezoelectric transducer (attached directly to the specimen surface), which converts recorded displacements into an electrical signal.

The nanoindenter device was equipped with a Physical Acoustics Corporation (PAC) WSα wide-band (100–1000 kHz) AE sensor, which showed a superior combination of frequency and sensitivity characteristics over other tested sensors (PAC Micro30S, PAC F15I-AST). The Zn single-crystal (with FIB-milled micropillars on the surface) was attached to the transducer over a layer of vacuum grease to ensure effective acoustic coupling. Mechanical bonding ('clipping') was carried out by means of a thin metallic strip bent over the sample and fixed at both ends to the device, ensuring a constant contact pressure during the compression tests. The recorded signal was amplified using the Vallen AEP5 pre-amplifier set to 40 dB_AE. Data acquisition and processing were performed using the computer-controlled Vallen AMSY-6 system. Data acquisition was carried out in continuous data streaming mode, i.e., the whole raw acoustic data sets were recorded for further post-processing at a sampling rate of 2.5 MHz.

*Identification of AE events.* To individualize the AE events, an in-house script implemented in Matlab was used. The threshold voltage was set to $V_{th} = 0.01$ mV, this value being slightly above the background noise. The hit definition time (HDT), i.e., the minimum period between two subsequent AE events, used for the separation of events was 20 µs.

In Supplementary Fig. 11 various parameters of a representative event related to the AE measurements are defined following the standard AE event individualization and parametrization procedure[44,46]. The original AE waveform $V(t)$ is plotted in the inset as a function of time $t$. The AE event energy is defined as the area under the squared signal amplitude curve:

$$E = \int_{t_b}^{t_e} V^2(t)\mathrm{d}t, \qquad (6)$$

where $t_b$ and $t_e$ denote the beginning and end of the event, respectively (that is, $E$ is the extent of the area shaded in blue in Supplementary Fig. 11). The AE counts are defined as the number of data points (in absolute values) crossing the threshold level $V_{th}$. The duration of one AE event is defined as the time between the first and the last AE count in that event.

*Data validation.* The common source of both load drops and AE events in the tested micropillars are dislocation avalanches in the basal plane. In order to exclude any other external effects that might lead to the generation of AE events, additional aspects of the AE measurement had to be addressed: (i) friction between the indenter's flat diamond tip and the top of the pillars and (ii) possible noise or vibrations from external sources and the nanotesting device itself.

To address point (i) we investigated six micropillars with identical geometry, fabricated by three different methods for this purpose. Two pillars were prepared with Pt coating, two with C coating and two pillars without any coating on top. Although three materials with different friction properties were used, the analysis produced practically identical results with respect to the AE events and strain bursts. To avoid the presence of any spurious extrinsic vibrations considered within point (ii), three further compression tests were carried out where a special tip suspension was applied––another elastic part (a piece of rubber) was added to the device to isolate possible vibrations and noises from external sources. Just as in the previous case (i), this analysis demonstrated that there were no observable differences in the AE data compared to the tests without this additional suspension.

*The overlapping of AE events.* Assuming that the AE events originate from individual well-defined plastic events (i.e., dislocation avalanches related to stress drops) and there are no significant scattering and echoing mechanisms during the wave propagation, one may expect an exponential decay of the waveform resulting from intrinsic absorption[47,48]. In that case, the relationship between the maximum squared amplitude

$$A^2 = \max_{t \in [t_b, t_e]} V^2(t), \qquad (7)$$

and the duration $T = t_e - t_b$ could be written as

$$A^2(T) = V_{th}^2 \exp\left(\frac{T}{\tau}\right), \qquad (8)$$

where $\tau$ is a timescale characterizing the rate of absorption[48]. This relation was fitted to all data points that were detected under the same noise conditions. This set of data contained more that 13,000 events from the compression tests on Zn pillars with various dimension. The data trends and exponential fits shown in Supplementary Fig. 12 prove the validity of relation (8); thus, we concluded that the majority of detected events are due to short pulse-like events at the source attenuated only by intrinsic absorption, while recording of wave reflections and overlapping events is not common with the AE event individualization parameters used in this study (see above). It is also noted, that the fitted value of $\tau = 45$ µs is below the typical timescales characteristic of the Omori-law and waiting time distributions in Fig. 3.

*Rates of aftershocks and foreshocks.* Large AE events, similarly to earthquakes, are usually followed by several aftershocks. To quantify the rate of these aftershocks the following procedure was implemented. First, we select an energy interval $[E_{ms} - \Delta E/2, E_{ms} + \Delta E/2]$ and consider only AE events with energies falling in this given bin. These will be the main shocks with energy $E_{ms}$. The sequence of events (aftershocks) corresponding to each main shock lasts until an event with energy falling in this or larger bin takes place. The time after the main shock $t$ is binned logarithmically, and the AE events in the sequence following the main shock falling in each bin are counted, and then repeated for all main shocks with energy $E_{ms}$. To obtain the rate of the aftershocks $r_{as}(t)$ the number of events in the time bin around $t$ is normalized with the bin width and also with the number of sequences that reached the given length $t$. The obtained $r_{as}(t)$ curves for $d = 32$ µm pillars are plotted in Fig. 3a, b. The corresponding figures for smaller micropillars are shown in Supplementary Fig. 4a–d.

In the case of foreshock rates $r_{fs}$ the same procedure was adopted and inverted in time to investigate sequences before main shocks. The obtained rates for $d = 32$ µm pillars are seen in Fig. 3c and for smaller ones in Supplementary Fig. 4e, f.

*Waiting time of AE events.* The waiting time distributions of Fig. 3d–f are obtained as follows. The identification of the individual AE events described above in section

'Identification of AE events' yields the time $t_i$ of each event. The waiting time is then simply $t_{w,i} = t_{i+1} - t_i$, and the distribution of these values is computed.

Since only those AE events can be detected that rise above the background noise, it is important to check the role of thresholding in the obtained distributions. To this end, the procedure described above was repeated after considering only events with energies $E$ larger than a threshold $E_{th}$. According to Supplementary Fig. 13a they only differ in the exponential tail characterized by parameter $t_0$ related to the average time between subsequent uncorrelated event clusters. As seen, increase of $E_{th}$ leads to fewer detected events and, thus, an increased $t_0$. To prove that thresholding does not influence the conclusions of the paper, in Supplementary Fig. 13b the distributions were re-scaled with the average waiting time $\langle t_w \rangle$ corresponding to the given threshold $E_{th}$. The obtained collapse of the curves means that $t_0 \propto \langle t_w \rangle$ similarly to what was obtained in the case of different platen velocities $v_p$ (Fig. 3e, f), and it proves scale-invariance of the AE events.

*Energetic considerations.* In this section background discussion is provided for the comparison of the energies of strain bursts and the corresponding AE events with an emphasis on the analogies with earthquakes. We start by noticing that there are three relevant energy quantities one may look at. The first one we call *injected energy* and is the work done by the compression device during an event:

$$E_{inj} = \bar{F}\Delta s \approx \frac{\sigma_0 + \sigma_1}{2} A\Delta s, \qquad (9)$$

where $\bar{F}$ is the average force exerted by the device, $\Delta s$ is the displacement of the punch tip, $\sigma_0$ and $\sigma_1$ are the stresses before and after the event and $A$ stands for the cross-section of the sample. This expression is equivalent to Eq. (3.14) in the review article on earthquake physics[49] where it is termed *potential energy*. If we assume a quick event with a stress drop of $\Delta\sigma = \sigma_0 - \sigma_1$, then

$$E_{inj} \approx \frac{A}{k}(\sigma_0 - \Delta\sigma/2)\Delta\sigma \propto (\sigma_0 - \Delta\sigma/2)\Delta\sigma, \qquad (10)$$

where $k$ is the stiffness of the device.

This work $E_{inj}$ is not necessarily equal with the dissipated energy, since the energy stored as elastic energy may change during an event. The second relevant energetic quantity is, therefore, the *dissipated energy* that actually equals the change of the elastic stored energy of the system (in this sense, the external work initially increases the elastic energy of the sample part of which then gets released due to plastic processes). Assuming a $v \propto \sigma$ linear relationship between the local stress and the drag acting on dislocations, the change in the elastic energy can be written in the form:

$$E_{dis} \propto \sum_{i=1}^{N} \int_{t_0}^{t_1} v_i^2 l_i \mathrm{d}t, \qquad (11)$$

where $t_0$ and $t_1$ mark the beginning and the end of an event, respectively, and $v_i$ and $l_i$ are the velocity and length of the $i$th dislocation or dislocation segment, respectively. This formula is the analogue of Eq. (3.12) of ref. [49], that is termed *radiated energy* for earthquakes.

The dissipated energy may create, e.g., heat or elastic waves. With an acoustic transducer some part of the energy released in the form of elastic waves can be detected (denoted by $E$ in Eq. (6)). Here we arrive at the issue of efficiency. It is, of course, not known what portion of the dissipated energy gets converted into elastic waves and what fraction of it can be measured at the surface. Here we make the simplest and most straightforward assumption that this ratio does not depend on the energy of the event, so, the measured AE energies are representative of the released energy: $E \propto E_{dis}$. We note that our situation is somewhat simpler than earthquakes since here only one type of deformation is active contrary to earthquakes where rupture and thermally activated processes may also play a role.

Concerning the relationship between $E_{inj}$ and $E_{dis}$ the natural assumption is again a linear dependency in the average sense. This idea, however, can be tested by the DDD simulations as there we have direct access to microstructural data, i.e., dislocation positions and velocities. It is evident from Fig. 4c in the main text that $E_{dis} \propto E_{inj}$ holds for large energies.

In the case of experiments only the injected energy $E_{inj}$ and the detected acoustic energy $E$ can be measured. In Fig. 2c and Supplementary Fig. 3 we plot the two quantities against each other for individual events. As seen, in this case a linear relationship between $E_{inj}$ and $E$ is obtained that seem to match the assumptions mentioned above.

## Simulations

*Discrete dislocation dynamics.* The model to be investigated is one of the simplest discrete dislocation systems that still incorporates the following fundamental physical properties of dislocations:

- $1/r$-type long-range interactions between dislocation lines.
- Non-conservative motion of dislocations due to the strong phonon drag.
- Geometrically constrained motion of dislocation lines, since at low temperatures they can only glide in certain planes (called glide planes). As a result, the system cannot reach a global energy minimum state, rather, it gets trapped in a meta-stable configuration.

The system consists of straight edge dislocations parallel with the $z$ axis, and their slip planes are parallel with the $xz$ plane (single slip). Since the system is translationally invariant along the $z$ axis it can be considered two-dimensional (2D) and it is sufficient to track the motion of dislocations in the $xy$ plane. In this set-up the Burgers vector points in the $x$ direction and, thus, reads as $\mathbf{b} = s(b, 0)$, where $s \in \{+1, -1\}$ is the sign of the dislocation, that can be understood as some kind of charge. Supplementary Fig. 14a shows an example of such a 2D dislocation configuration. The colours of dislocations represent their sign and the background colour refers to the local shear stress within the embedding elastic medium.

Because of the strong dissipation due to phonon drag, the motion of dislocations is assumed to be overdamped, that is, the force acting on a dislocation of unit length is proportional to its velocity. If the system consists of $N$ dislocations and $\mathbf{r}_i = (x_i, y_i)$ denotes the position of the $i$-th ($i = 1, \dots, N$) dislocation then the equation of motion reads as

$$\dot{x}_i = M s_i b \left[ \sum_{j=1; j\neq i}^{N} s_j \sigma_{\mathrm{ind}}(\mathbf{r}_i - \mathbf{r}_j) + \sigma \right], \quad (12)$$

$$\dot{y}_i = 0. \quad (13)$$

Here $M$ is the dislocation mobility, $\sigma$ is the externally applied shear stress and $\sigma_{\mathrm{ind}}$ is the shear stress field generated by individual dislocations. For the latter the solution corresponding to isotropic continua is used[50]:

$$\sigma_{\mathrm{ind}}(\mathbf{r}) = \frac{\mu b}{2\pi(1-\nu)} \frac{x(x^2 - y^2)}{(x^2 + y^2)^2}, \quad (14)$$

where $\mu$ and $\nu$ denotes the shear modulus and the Poisson number, respectively. Dislocations are arranged in a square-shaped simulation area and periodic boundary conditions (PBC) are applied. The emerging image dislocations alter the stress field of Eq. (14) (that corresponds to an infinite medium), which can be obtained using a Fourier method (see Supplementary Fig. 14b)[51]. The equations of motion ((12), (13)) are solved using a fully implicit scheme that makes usage of annihilation unnecessary, so, it is not implemented[52].

With the application of the PBCs surface effects that may be important for small scale samples are neglected in the simulations. Indeed, in nanopillars it was found that exhaustion hardening[53] as well as dislocation source truncation[54] represent key physics in the plastic deformation that lead to size-effects. These single-dislocation properties are important at scales comparable or smaller than the average dislocation spacing, being around $\rho^{-0.5} \approx 0.1\ \mu m$. In our experiments the micropillars are rather large (8–32 $\mu m$) compared to previous studies, and at this scale collective dislocation dynamics is expected to dominate plasticity and boundary effects can be neglected. This explains our choice for the PBCs that allowed us to concentrate on collective dislocation phenomena. It is also noted, that in experiments plastic deformation leads to the shape change of the sample and may cause some lattice rotation. These effects are not expected to have a significant role in the observed dynamics and are neglected in the simulations.

One of the main advantages of the model system introduced is that the dislocation interactions exhibit a $1/r$-type decay. This means that apart from the average dislocation spacing (being equal to $\rho^{-0.5}$, where $\rho$ is the total dislocation density) no additional length scales appear in the model. One may, thus, introduce dimensionless variables by measuring length, stress, strain and time in units summarized in Supplementary Table 1, where notation $G = \mu/[2\pi(1-\nu)]$ is used.

Initially, an equal number of positive and negative sign dislocations are positioned randomly in the square-shaped simulation area with uniform distribution. At zero applied stress the system is first let to evolve into a relaxed equilibrium configuration. After that the applied shear stress is increased using a protocol emulating the experimental set-up of micropillar compression. Namely, the applied stress is computed at every time step according to

$$\sigma' = r'(v'_{\mathrm{p}} t' - \varepsilon' L'), \quad (15)$$

where $v'_{\mathrm{p}}$ is the platen velocity (see Fig. 4a), $t'$ is the simulation time, $r'$ is a constant characterizing the strength of the spring connecting the platen and the dislocation system, and $\varepsilon'$ is the accumulated plastic shear strain computed as:

$$\varepsilon'(t') = \sum_{i=1}^{N} s_i [x'_i(t') - x'_i(0)]. \quad (16)$$

In the simulations $r' = 1/32$ was used and the platen velocity (if not stated otherwise) was set to $v'_{\mathrm{p}} = 1.6 \times 10^{-4}$.

*Event detection.* The overdamped dynamics used in Eqs. ((12), (13)) reflects the fact that dislocation motion is a highly dissipative process during which stored elastic energy $E'_{\mathrm{el}}$ of the embedding crystal is transformed into other types of energies (e.g., heat or elastic waves). This energy dissipation rate $r'_{\mathrm{en}}$ can be obtained as

$$r'_{\mathrm{en}} = -\dot{E}'_{\mathrm{el}} = \sum_{i=1}^{N} (v'_i)^2, \quad (17)$$

where $v'_i = \dot{x}'_i$ is the velocity of the $i$th dislocation.

Stress drop detection is based on the finding that in active periods the dissipation rate $r'_{\mathrm{en}}$ increases several orders of magnitudes as demonstrated on an exemplary event in Supplementary Fig. 15. To obtain the beginning $t'_{\mathrm{b}}$ and end $t'_{\mathrm{e}}$ of

the event a threshold of $r'_{\mathrm{th}} = 5 \cdot 10^{-6}$ was used for the dissipation rate as demonstrated in Supplementary Fig. 15. The size of the stress drop then follows as $\Delta\sigma' = \sigma'(t'_{\mathrm{e}}) - \sigma'(t'_{\mathrm{b}})$.

As seen in Supplementary Fig. 15, a plastic event exhibits a fine structure with many peaks in the dissipation rate $r'_{\mathrm{en}}$. In order to emulate an AE detector, an additional threshold $r'_{\mathrm{th,AE}}$ is defined that characterises the sensitivity of the detector: whenever $r'_{\mathrm{en}} > r'_{\mathrm{th,AE}}$ the detector is able to measure the dissipation rate. With this, emulated AE events can be defined as demonstrated in the inset of Supplementary Fig. 15. The threshold $r'_{\mathrm{th,AE}}$ breaks up the signal in individual AE events, with their energy $E'$ being the size of the area shaded alternately in blue and red colour. Data processing was carried out with the utilization of the NumPy library[55].

From the list of stress drops and AE events the AE count rate, the correlation between stress drops and AE energies, the aftershock rates and the waiting time distributions in Fig. 4b–e were determined with the same procedure as for experiments. The role of the threshold $r'_{\mathrm{th,AE}}$ used to model AE detector sensitivity was also investigated. According to Supplementary Fig. 16 the Omori law as well as the productivity law are recovered in a wide range of thresholds, however, small thresholds lead to the coalescence of events leading to a deviation from the power-law behaviour for small times $t'$. In Fig. 4d–e $r'_{\mathrm{th,AE}} = 3.16$ was used for AE individualization.

## Data availability

Experimental data generated in this study have been deposited in the Zenodo database at https://doi.org/10.5281/zenodo.5897653.

## Code availability

The numerical methodologies used in this study can be built with the instructions in the 'Methods' section. The codes for numerical simulations are also available in the Zenodo database at https://doi.org/10.5281/zenodo.5904306.

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

## Acknowledgements

The work was supported by the Hungarian Ministry of Human Capacities within the ELTE Institutional Excellence Program TKP2020-IKA-05 (P.D.I, D.U, G.P., D.T., Z.D. and I.G.). Financial support from the National Research, Development and Innovation Fund of Hungary is also acknowledged (contract numbers NKFIH-K-119561 and NKFIH-FK-138975; P.D.I, D.U., G.P., D.T. and I.G.). Funding from ÚNKP-20-3, ÚNKP-21-4 (D.U.), and ÚNKP-21-3 (G.P.) in the framework of the National Excellence Program of the Ministry for Innovation and Technology from the source of the National Research, Development and Innovation Fund is also acknowledged. This work was also funded by the Czech Science Foundation (grant No. 19-22604S; M.K. and F.C.).

## Author contributions

P.D.I. designed the research and supervised the project. P.D.I., D.U., Z.D., D.T. and I.G. designed and developed the microdeformation stage. D.U. performed micropillar fabrication and compression experiments as well as EBSD and X-ray measurements. D.U., M.K., K.M. and F.C. performed the AE measurements. S.K. assisted with the sample preparation and analysis. P.D.I., D.U. and S.K. analysed the experimental data. I.G. assisted at every experimental measurement. G.P. developed and performed the simulations and performed the slip band analysis. P.D.I., D.U., G.P., S.K., M.K. and K.M. wrote the paper, with contributions from all authors.

## Funding

## Competing interests

The authors declare no competing interests.
