## [Peer Review File · Nature Communications]

Dislocation avalanches are like earthquakes on the micron scaleReviewers' Comments:

Reviewer #1:

Remarks to the Author:

Manuscript „Dislocation Avalanches: Earthquakes on the Micron Scale“ by P.D. Ispánovity et al.

The manuscript deals with the acoustic emission occurring during deformation of hexagonal metal (Zn) single crystal micropillars and relates these emissions with the acoustic emission of earthquakes. In terms of the signal types, there are undoubtedly strong similarities – beside the different amplitudes and time regimes of course. Such similarity was already stated earlier in principle, but not proven on basis of a quantitative evaluation like in the present case. The compression set-up used for the micropillar experiments was specially developed for complementary SEM studies. Moreover, the underlying dislocation movement was modelled by means of discrete dislocation dynamics simulation. In summary, the manuscript is a very interesting study worth being published in Nature. However, some clarifications are necessary:

- i) In order to determine dislocation densities on basis of X-ray investigations, the contrast factors have to be known. In particular, there is only limited information on contrast factors for hexagonal metals. It is unclear, however, which contrast factors were used. Is there a reference on preliminary work or on contrast factors in literature?
- ii) The high-resolution EBSD is not explained to sufficient detail. The authors should give some more information on these measurements.
- iii) The energy of single acoustic emission events seems to be calculated on basis of the time domain. The authors should give the reason why they didn't use the frequency domain as some other authors did in literature.

Reviewer #2:

Remarks to the Author:

Advantages:

The dislocation avalanches resulted in strain burst in micropillar is well known crystal plasticity experiment in the last two decades. Focused ion beam (FIB) operation is used to fabricate square-based pillars under compression force. The authors have set up an ingenious experiment facility in this paper to measure the signal of dislocation avalanches (AE) events. Then, there is a nice creativity topic to analogy AE on the micron scale to earthquakes in macro scale. Achieved from this work is that the realization of nontrivial task of detecting extremely weak AE waves which arise during micropillar deformation. The principle of the emission of acoustic waves in materials is analogous to earthquakes. Plastic deformation is caused by the local rearrangement of dislocation lines in a crystal. This process is strongly dissipative and part of the released elastic energy escapes in the form of elastic waves, that can be detected at the surface. The complex dynamic behavior observed in experiments is the result of the spatio-temporal correlations of the dislocations due to their long-range elastic interactions mechanism.

Disadvantages:

1. Zn single crystal with hexagonal closed-packed (HCP) structure is used in the experiment. While, Al Co and Ni FCC single crystals are used in the most experiments of micron pillar compression. The question is that these metals can be used by analogous with earthquake like Zn expressed in this paper.
2. Is there small or finite deformation used in DDD simulation? For my knowledge, there is no pure DDD program which can compute the finite deformation of crystal. The finite deformation with shear band can be found from experiment in Figures. For this configuration, the spin rate of plastic deformation should be considered. There is no big influence on AE signal, which is dominated by dislocations avalanche. However, it has some effect on plastic energy in micron pillar, which is analogy to earthquake.

Reviewer #3:

Remarks to the Author:

This very interesting manuscript proposes an analysis of plastic avalanches in compressed Zn micropillars from a combined analysis of stress drops and AE. Although the characteristics of the underlying dynamics revealed from this work (i.e. scaling laws, time correlations) are not new, this combination of methods, which is new to my knowledge, allows to reconcile AE data of dislocation avalanches obtained at bulk scales in one hand, and intermittency of plasticity observed directly from loading curves at μm - to sub- μm scales. It also reveals an interesting 2-levels structure of brief instabilities (measured from AE) inserted into a single stress drop, considered as a single avalanche in classical micropillar analyses. The experimental work is nice and looks sound. For these reasons, I think that this work is worth publishing.

On the other hand, the authors argue, already from the title, about a very strong analogy between dislocation avalanches and earthquakes. I am clearly less enthusiastic about this claim, for several reasons, explained in more details below: the observed scaling laws (power law distributions of AE energies, time correlations between AE events, productivity law) have been already reported in the literature; the time correlations analyzed here cannot be compared directly to aftershock analyses in earthquake studies, as there is no information about the location of events, or about spatio-temporal correlations; and, finally, the "foreshock" rates are close to the "aftershock" rates, in striking contrast with earthquakes. For these reasons, I would suggest to significantly understate this claim in the paper, and instead maybe focus more on some others interesting points (some are listed below).

1. L25-27: this is not always the case, and depends on the material (see ref. 28).

2. L65-66, and more generally about the evolution of stress and strain during an "avalanche". As shown on Extended Data Fig 6, the characteristics of the loading device implies that an avalanche is characterized by both a stress drop $\Delta\sigma$ and a strain increment, which is classical for such type of experiment on micropillars. Consequently, the stress drop is maybe not the best measure of the strain energy released during the avalanche. An estimate of this strain energy should combine the evolution of both stress and strain. If the slope of the descending parts of the curves of Extended data Fig 6 is constant, and is simply a measure of the stiffness of the machine, then the released energy should scale as $\Delta\sigma^2$. In any case, on figure 2, it might be more judicious to discuss the scaling between AE energy and this strain energy, not the stress drop itself. On this figure 2, the authors report a scaling $E_{\text{AE}} \sim [\Delta\sigma]^{1.2}$, which would mean $E_{\text{AE}} \sim (\Delta\sigma^2)^{0.6}$. In other words, this would suggest that the AE energy scales with, but is not proportional to, the released strain energy. This differs from what is generally assumed to interpret AE data in the lab. This is reminiscent of the debate about seismic efficiency of earthquakes (e.g. [Kanamori and Brodsky, 2004]). This point is essential for a comparison of dislocation avalanches as observed from AE, or directly from stress-strain curves in μm - to sub μm -systems. It would therefore merit, in my opinion, a more thorough investigation from this novel dataset (maybe complemented with DDD simulations ?).

3. L75-77 and Figures 1c to 1e: these key figures are too small, and so it is difficult for the reader to appreciate this discussion. I might suggest to separate these figures from figure 1a. It might be also useful to represent as well the AE activity vs the strain (not time); see comment #2 above.

4. Figure 3, and aftershock analysis. The results reported on figure 3 are a clear signature of time correlations in dislocations dynamics. Such type of correlations were already characterized in another hexagonal material at bulk scale [Weiss and Miguel, 2004]. However, correlation is distinct from causality, i.e., in this context, from mechanical triggering. In case of earthquakes, the causal structure can be inferred from seismic data using a specific methodology (see [Marsan and Lengliné, 2008] or [Houdoux et al., 2021]). However, this requires an information about the location of the slip events, which is not available here. It is therefore difficult to discuss these results in terms of "aftershock triggering" *stricto sensu*. In addition, a comparison of figures 3a and 3c show a rather symmetric

pattern, i.e. a “foreshock rate” close to the “aftershock rate”. This is fundamentally different from earthquakes. This might result from the ambiguity to define “mainshocks” from the present data. In conclusion, these data reveal time correlations in dislocation dynamics, characterized by a power law decay, up to a time scale of the order of the duration of stress drops, i.e. “avalanches” defined from the stress-strain curves.

Concerning this power law decay, another point should be scrutinized: owing to the loading apparatus (and its intrinsic stiffness) as well as the finite sample size, the triggering of an avalanche implies a stress drop during the avalanches (as observed here). Considering that, at the local level, the stress controls the unpinning of dislocations as well as their velocity, this decaying stress must have an impact on the rate of dislocation activity during such a stress drop. In other words, there is a coupling between dislocation avalanches and the dynamics of the global elastic system, and it would be useful to analyze the potential impact of this coupling on the time correlations reported above.

More generally, these results reveal a two-levels structure of short bursts (measured from AE and correlated in time) embedded within larger instabilities leading to system-size stress drops. There is a clear link between large AE rates and these large instabilities, both AE bursts and large avalanches are power-law distributed, while AE bursts are correlated in time, but not the large instabilities. All of this is potentially of great interest for our understanding of dislocation dynamics in general, more particularly in μm - to sub- μm systems, and for the interpretation of micropillar compression tests. In my opinion, this would deserve a longer discussion in this manuscript.

5. Figures 3e and 3f. These figures suggest that AE events are correlated in applied strain, not time. A similar problem was recently explored for the deformation of granular media [Houdoux et al., 2021]. This distinction is not just semantic; it might be important in terms of underlying physics (in other words, does time really matter here ?).

6. Modelling: The DDD simulations are run using periodic boundary conditions. However, in μm to sub- μm pillars, finite size effects are likely playing an important role, changing the nature of dislocation sources, of nucleation mechanisms, ect.. A comment on this would be welcome.

7. L204-205: OK for HCP materials. This might be quite different for FCC or BCC materials, for which short-range mechanisms are expected to play a significant role.

8. L211-212: “and most likely crystalline metals/solids in general”. I disagree on this statement. FCC and BCC metals do not exhibit such scale-free dynamics in general [Weiss et al., 2021; Zhang et al., 2017]

9. L214: spatial correlations were not analyzed here.

10. L302: Could the authors give the corresponding applied strain-rate ?

11. Section on “Edge detection”. This section describes a nice example showing the activation of a single slip band associated to a single stress drop (see also Fig 1). It is however not clear whether the authors performed such image analysis at a regular periodicity ? or just for few examples of stress drops.

12. Extended figure 13: it would nice to show the evolution of the stress as well here.

Refs.:

Houdoux, D., A. Amon, D. Marsan, J. Weiss, and J. Crassous (2021), Micro-slips in an experimental granular shear band replicate the spatiotemporal characteristics of natural earthquakes, *Communications Earth & Environment*, 2(1), 1-11.

Kanamori, H., and E. E. Brodsky (2004), The physics of earthquakes, *Reports on Progress in Physics*, 67(8), 1429.

Marsan, D., and O. Lengliné (2008), Extending earthquakes' reach through cascading, *Science*, 319, 1076-1079.

Weiss, J., and M.-C. Miguel (2004), Dislocation avalanche correlations, *Mat. Sci. Eng. A*, 387-389(292-296).

Weiss, J., P. Zhang, O. U. Salman, G. Liu, and L. Truskinovsky (2021), Fluctuations in crystalline plasticity, *Comptes Rendus. Physique*, 22(S3), 1-37.

Zhang, P., O. G. Salman, J. Y. Zhang, G. Liu, J. Weiss, L. Truskinovsky, and J. Sun (2017), Taming intermittent plasticity at small length scales, *Acta Materialia*, 128, 351-364.

Authors' Response to Reviews of

Dislocation Avalanches: Earthquakes on the Micron Scale

Péter Dusán Ispánovity, Dávid Ugi, Gábor Péterffy, Michal Knappek, Szilvia Kalácska, Dániel Tüzes, Zoltán Dankházi, Kristián Máthis, František Chmelík, István Groma
Nature Communications, NCOMMS-21-22888-T

RC: *Reviewers' Comment*, AR: Authors' Response, Manuscript Text

WE THANK THE REVIEWERS FOR THEIR VALUABLE THOUGHTS AND COMMENTS ON THIS WORK, AND FOR RECOMMENDING THE MANUSCRIPT FOR PUBLICATION. OUR RESPONSE IS PROVIDED BELOW. CHANGES TO THE ORIGINAL MANUSCRIPT HAVE BEEN HIGHLIGHTED (~~DELETED PARTS~~ , ADDED PARTS) IN THIS DOCUMENT AND TRACED BY "LATEXDIFF" IN THE REVISED FILE.

Reviewer #1

RC: *Manuscript „Dislocation Avalanches: Earthquakes on the Micron Scale“ by P.D. Ispánovity et al. The manuscript deals with the acoustic emission occurring during deformation of hexagonal metal (Zn) single crystal micropillars and relates these emissions with the acoustic emission of earthquakes. In terms of the signal types, there are undoubtedly strong similarities – beside the different amplitudes and time regimes of course. Such similarity was already stated earlier in principle, but not proven on basis of a quantitative evaluation like in the present case. The compression set-up used for the micropillar experiments was specially developed for complementary SEM studies. Moreover, the underlying dislocation movement was modelled by means of discrete dislocation dynamics simulation. In summary, the manuscript is a very interesting study worth being published in Nature. However, some clarifications are necessary:*

AR: We thank the Reviewer the careful assessment and positive evaluation of our work and for acknowledging the main findings of our paper.

RC: *i) In order to determine dislocation densities on basis of X-ray investigations, the contrast factors have to be known. In particular, there is only limited information on contrast factors for hexagonal metals. It is unclear, however, which contrast factors were used. Is there a reference on preliminary work or on contrast factors in literature?*

AR: We thank the Reviewer for pointing this out. Here we clarify how the contrast factors were determined. We start again from the intensity distribution generated by dislocations that was shown by Groma *et al.* [1] to decay asymptotically as

$$I(q) = \frac{\Lambda}{4\pi^2} \langle \rho \rangle \frac{1}{q^3}, \quad (1)$$

where $\langle \rho \rangle$ is the dislocation density, Λ is a constant depending on the diffraction, the Burgers, and the line direction vectors of the dislocation (see below), and $q = \frac{2}{\lambda} [\sin(\Theta) - \sin(\Theta_0)]$, in which λ is the wavelength of the X-ray, Θ is the scattering angle, and Θ_0 is the Bragg angle of the reflection used. The quantity Λ is commonly written in the form $\Lambda = \frac{\pi}{2} g^2 b^2 C_g$ [1, 2, 3], where C_g is called the “contrast factor”.

In order to determine Λ we use linear elasticity, i.e., it is assumed that dislocations are embedded in an elastic medium. Therefore, large deformations around the dislocation core are neglected, as due to their small

total volume they practically do not contribute to the diffraction. So, the type of crystal structure enters the calculation through the elastic constants of the medium, see below. As the first step, one has to calculate the integral

$$\phi_{ijkl} = \int_0^{2\pi} \partial_i u_j(r, \varphi) \partial_k u_l(r, \varphi) r^2 d\varphi \quad (2)$$

where $u_i(r, \varphi)$ is the displacement field generated by a straight dislocation of a specific type that is parallel to the z axis and positioned at the origin of the xy plane. The variables r and φ denote the polar coordinates in the xy plane. (It should be noted that due to the $1/r$ decay of the distortion generated by a dislocation the above integral is independent of r).

From Eq. (2) the contrast factor for a given dislocation type and diffraction vector \mathbf{g}

$$C_{\mathbf{g}} = \frac{2\pi^2}{b^2 g^4} \phi_{ijkl} g_i g_j g_k g_l \quad (3)$$

where the diffraction vector \mathbf{g} is given in the coordinate system corresponding to \mathbf{u} used in Eq. (2) [2, 3].

For isotropic materials there is a closed expression for C [1], but in the case of an elastically anisotropic material it can only be determined numerically.

We now shortly summarize the main steps of the calculation. According to [4] the displacement field can be given as

$$u_i = \frac{1}{\pi} \sum_{\alpha=1}^3 A_{i\alpha} D_{\alpha} \ln(x_1 + p_{\alpha} x_2) \quad (4)$$

where p_{α} are the roots of the sextic equation provided by the elastic equilibrium equation (see Eq. (10.9) in [4]). The values of p_{α} can, thus, be determined only numerically. $A_{i\alpha}$ and D_{α} are functions of the roots p_{α} and the elastic constants [4]. With the knowledge of the parameters in Eq. (4) corresponding to the dislocation considered, the value of C can be determined in a straightforward manner.

It should be noted, that the calculation of the contrast factor requires the solution of an elasticity problem. So, as said above, the actual crystalline structure does not make a real difference. As a consequence the calculation can be performed with a similar method for FCC, BCC or HCP structures as well. What really makes a difference is the relative strength of the elastic anisotropy.

In most cases there are several types of dislocations present or created during the deformation. In this case one has to determine the appropriate weighted average Λ . Since the relative populations of the different types of dislocations are often unknown, one has to measure as many reflections as possible. After determining the $\rho^* = \Lambda\rho$ values from the different peaks the relative populations of the different dislocations can be determined. If, for some reason, only a few reflections can be measured one has to make some assumption about the dislocation populations.

In the investigations reported in this paper X-ray line profile analysis was applied to determine the initial dislocation density of the samples. For simplicity, we assumed that each type of dislocations with Burgers vector in the basal plane are present in equal number. Dislocations with non-basal Burgers vector have larger energy, so they could be neglected. In the measurements the diffraction peak corresponding to $(1, 0, \bar{1}, 1)$ was used. The corresponding average contrast factor (determined by the program ANIZC at <http://metal.elte.hu/anizc/program-hexagonal.html>) is $C = 0.1992$, while $g^2 b^2 = 2.54$. From this $\Lambda = \pi g^2 b^2 C / 2 = 0.506$ [5, 6].

The shortened version of this discussion was added to the corresponding part of Methods as a new paragraph:

For this reason for the determination of the initial dislocation density one has to make assumptions about the relative densities of dislocations of different types. Since the energy of dislocations with Burgers vector lying in the basal plane is lower compared to other types it is reasonable to assume that in the original undeformed sample each slip system with a Burgers vector lying in the basal plane is equally populated. To account for elastic anisotropy the corresponding average Λ was determined numerically by the ANIZC program (<http://metal.elte.hu/anizc/program-hexagonal.html>) using the elastic moduli of Zn, yielding $\Lambda = 0.506$ [5] [6].

RC: *ii) The high-resolution EBSD is not explained to sufficient detail. The authors should give some more information on these measurements.*

AR: We thank the Reviewer for this comment. Indeed, the details of the HR-EBSD evaluation were lacking, therefore the following text was added to the manuscript. We also show here an exemplary diffraction pattern and the 20 regions of interest chosen for the analysis (Figure 1a) and the resulting GND density map (Figure 1b).

To calculate the initial geometrically necessary dislocation (GND) density, a digital image cross-correlation based technique called high (angular) resolution electron backscatter diffraction (HR-EBSD) was applied [7]. HR-EBSD determines local strain and stress tensor components with the help of the raw diffraction patterns (for details see [7]). This method requires a reference diffraction pattern for the image correlation, that is ideally captured in the strain-free state of the lattice. A perfect reference pattern is often difficult to obtain experimentally, therefore in our case a pattern with the presumably lowest stress is chosen, creating a relative scale for the GND density. Diffraction patterns were recorded with approx. 500×500 px² resolution from an area of 16.2×14.2 μm^2 with a step size of 100 nm. The evaluation was carried out by BLGVantage CrossCourt v4.2 software. 20 regions of interest (of 128×128 px² each, Ext. Data Fig. 1) were selected from each diffraction pattern to carry out the HR processing with applied high and low pass filtering. All points in the map were evaluated. The estimated average value of $\rho^{\text{GND}} = 1.2 \times 10^{13} \text{ m}^{-2}$ was measured on a surface prepared by the same FIB conditions (30 kV, 3 nA) as it was used for the pillar fabrication prior to deformation. This value is close to the detection limit of the GND density by HR-EBSD, hence it is concluded that the sample preparation did not introduce a significant/measurable dislocation content in the sample.

RC: *iii) The energy of single acoustic emission events seems to be calculated on basis of the time domain. The authors should give the reason why they didn't use the frequency domain as some other authors did in literature.*

AR: We thank the Reviewer for this query and in the following we would like to elaborate more on the justification of employment of the time domain in our AE analyses. In this work, the time evolution of three different (yet complementary) data sets is of paramount interest, namely (i) the stress-time deformation curves, (ii) the sequence of SEM micrographs, and (iii) the AE response. In order to be able to correlate the AE signatures with the progression of the other two quantities, a standard definition of time-localized AE events and conventional event parameters were used according to Refs. [8, 9, 10] (see also Extended Data Figure 11). The AE energy of a particular AE event is thus calculated in the most direct way from the signal time series as $E = \int_{t_b}^{t_e} V^2(t)dt$ (Eq. 6), where $V(t)$ is the AE waveform as a function of time t , and t_b and t_e denote the

Figure 1: **a**, A typical Kikuchi pattern collected from the sample used for the HR-EBSD evaluation. 20 regions of interest are marked with yellow squares. **b**, The resulting GND density map. Black point (in the middle) marks the reference pixel.

beginning and end of the event, respectively. Another point in favor of this definition is that the recorded AE signals are burst-like and well-defined in time, i.e. not overlapping (also refer to Extended Data Figure 12).

By converting the AE waveforms corresponding to individual AE events into the frequency domain, the AE energy is, roughly, conserved. However, due to signal discretization and finite AE event duration, the Parseval's theorem

$$\int_{-\infty}^{\infty} V^2(t)dt = \frac{1}{2\pi} \int_{-\pi}^{\pi} \hat{V}^2(\omega)d\omega,$$

where $\hat{V}^2(\omega)$ is the Fourier transform of V^2 and ω is the angular frequency, does not hold. In other words, the equivalence between the time and frequency representation of the signal is not exactly maintained and certain errors in energy calculations will be introduced. For these reasons, we contend that here the calculation of AE event energy in the time domain is more convenient and precise.

On the other hand, we agree that the representation of AE waveforms in the frequency or time-frequency domains can provide additional information on the AE signal frequency spectra, reflecting the deformation dynamics characteristics. Yet, we believe that such analyses are beyond the scope and objectives of this paper.

Based on the above-elaborated considerations, we added the following text and references in the 'AE measurements' section:

In Extended Data Fig. 11 various parameters of a representative event related to the AE measurements are defined following the standard AE event individualization and parametrization procedure [8] [9]. The original AE waveform $V(t)$ is plotted in the inset as a function of time t . The AE event energy is defined as...

Reviewer #2

RC: *Advantages: The dislocation avalanches resulted in strain burst in micropillar is well known crystal plasticity experiment in the last two decades. Focused ion beam (FIB) operation is used to fabricate square-based pillars under compression force. The authors have set up an ingenious experiment facility in this paper to measure the signal of dislocation avalanches (AE) events. Then, there is a nice creativity topic to analogy AE on the micron scale to earthquakes in macro scale. Achieved from this work is that the realization of nontrivial task of detecting extremely weak AE waves which arise during micropillar deformation. The principle of the emission of acoustic waves in materials is analogous to earthquakes. Plastic deformation is caused by the local rearrangement of dislocation lines in a crystal. This process is strongly dissipative and part of the released elastic energy escapes in the form of elastic waves, that can be detected at the surface. The complex dynamic behavior observed in experiments is the result of the spatio-temporal correlations of the dislocations due to their long-range elastic interactions mechanism.*

AR: We thank the Reviewer the careful assessment and positive evaluation of our work and for acknowledging the main findings of our paper.

RC: *Disadvantages: 1.Zn single crystal with hexagonal closed-packed (HCP) structure is used in the experiment. While, Al Co and Ni FCC single crystals are used in the most experiments of micron pillar compression. The question is that these metals can be used by analogous with earthquake like Zn expressed in this paper.*

AR: The Referee is absolutely right in suggesting that the lattice structure may have an influence on the observed behaviour. The AE experiments reported here were so far only conducted for Zn, so we can only speculate about the case of FCC microcrystals. The reason why we used Zn is simply that bulk HCP samples typically emit much stronger AE signals than those with FCC or BCC structure. We believe that the crystal structure is important in an indirect way, by affecting the arrangement and possible reactions of dislocations. In HCP Zn studied here, dislocations with the lowest energy are those on the basal plane and, similarly to FCC, these dissociate into partials [11], so in the initial state the slip systems of this plane are expected to contain much larger amount of dislocations than the prismatic and pyramidal slip systems. Since the samples are oriented for basal slip, this becomes even more the case during compression, so, the emergent deformation is single slip with practically no forest dislocations on the other slip systems. This leads to the absence of short-range dislocation reactions and, thus, long-range forces dominate the dynamics. In FCC, on the other hand, due to the higher symmetry of the crystal we expect dislocations to be present on inclined planes, too, even if the sample is oriented for single slip. This would lead to short-range interactions that could dominate over the long-range interactions and perhaps change the observed behaviour. We extended the corresponding discussion in the manuscript, as seen below. Note, that a similar comment was also raised by Referee #3, see our response below.

The experiments and simulations reported here prove this long-standing hypothesis and reinforce that intermittency and scale-invariance characterizing plastic deformation of HCP single crystals (~~and, most likely, crystalline metals/solids in general~~) are related to the self-organized critical (SOC) behaviour of dislocations. In addition, we showed that plastic events, similarly to earthquakes, do not only exhibit spatial but also temporal clustering with long-range correlations, however, the involved length and timescales are profoundly different, as summarized in Extended Data Table 2. This phenomenon also raises analogy with many other physical systems exhibiting crackling noise [12]. It is known, however, that SOC behaviour is not ubiquitous in crystal plasticity, for instance, it is suppressed in materials

with FCC and BCC crystal structure and under multiple slip conditions likely because of short-range interactions related to dislocation reactions and also at high temperatures [13, 14, 15].

RC: *2. Is there small or finite deformation used in DDD simulation? For my knowledge, there is no pure DDD program which can compute the finite deformation of crystal. The finite deformation with shear band can be found from experiment in Figures. For this configuration, the strain rate of plastic deformation should be considered. There is no big influence on AE signal, which is dominated by dislocations avalanche. However, it has some effect on plastic energy in micron pillar, which is analogy to earthquake.*

AR: We thank the Referee for this interesting comment. Indeed, during the compression of the micropillars the change of the shape of the pillar does influence the stress field inside the specimen that may have an effect on the avalanche dynamics. The lattice rotation effect during compression may also play a role by changing the Schmid factors of the slip systems. We agree that this effect is probably weak in our case since the maximum deformation is not very high, the distribution of slip planes seem even along the height of the micropillar and we don't see noticeable changes in the statistical properties of AE events or strain burst during the compression.

In the simulations we apply periodic boundary conditions (see also our response to comment #6 of the 3rd Referee). In this case an infinite crystal is mimicked. As such, in this model the shape of the surface does not play a role and lattice rotation does not take place even at large deformation. As explained in the manuscript, the reason for using this type of generic simulations is that we focus on the role of long-range interactions that are believed to play the key role in the phenomena reported in this paper. If we intended to use a more detailed 3D DDD simulation of micropillar compression, the effects mentioned by the Referee might be of importance.

A new paragraph was added to section 'Discrete dislocation dynamics' in the Methods that also deals with this issue:

With the application of the PBCs surface effects that may be important for small scale samples are neglected in the simulations. Indeed, in nanopillars it was found that exhaustion hardening [16] as well as dislocation source truncation [17] represent key physics in the plastic deformation that lead to size-effects. These single-dislocation properties are important at scales comparable or smaller than the average dislocation spacing, in our case being around $\rho^{-0.5} \approx 0.1 \mu\text{m}$. In our experiments the micropillars are rather large ($8 - 32 \mu\text{m}$) compared to previous studies, and at this scale collective dislocation dynamics is expected to dominate plasticity and boundary effects can be neglected. This explains our choice for the PBCs that allowed us to concentrate on collective dislocation phenomena. It is also noted, that in experiments plastic deformation leads to the shape change of the sample and may cause some lattice rotation. These effects are not expected to have a significant role in the observed dynamics and are neglected in the simulations.

Reviewer #3

RC: *This very interesting manuscript proposes an analysis of plastic avalanches in compressed Zn micropillars from a combined analysis of stress drops and AE. Although the characteristics of the underlying dynamics revealed from this work (i.e. scaling laws, time correlations) are not new, this combination of methods, which is new to my knowledge, allows to reconcile AE data of dislocation avalanches obtained at bulk scales in one hand, and intermittency of plasticity observed directly from loading curves at μm - to*

sub- μm scales. It also reveals an interesting 2-levels structure of brief instabilities (measured from AE) inserted into a single stress drop, considered as a single avalanche in classical micropillar analyses. The experimental work is nice and looks sound. For these reasons, I think that this work is worth publishing. On the other hand, the authors argue, already from the title, about a very strong analogy between dislocation avalanches and earthquakes. I am clearly less enthusiastic about this claim, for several reasons, explained in more details below: the observed scaling laws (power law distributions of AE energies, time correlations between AE events, productivity law) have been already reported in the literature; the time correlations analyzed here cannot be compared directly to aftershock analyses in earthquake studies, as there is no information about the location of events, or about spatio-temporal correlations; and, finally, the “foreshock” rates are close to the “aftershock” rates, in striking contrast with earthquakes. For these reasons, I would suggest to significantly understate this claim in the paper, and instead maybe focus more on some others interesting points (some are listed below).

AR: We thank the Reviewer the careful assessment and positive evaluation of our work and for acknowledging the main findings of our paper. We absolutely agree with most of the statements of the Reviewer, however there are two points, where we would like to add a few comments:

- The Reviewer claims that the observed scaling laws were already reported in the literature. Indeed, we have referenced these works performed on bulk samples in the original manuscript. However, the current set-up allowed us to measure the Omori and productivity laws to much higher precision, to undoubtedly conclude about the power-law behaviour and to obtain the values of the scaling exponents. We believe that these are significant advancements to prior experiments from the perspective of our physical understanding of dislocation avalanches.
- The Reviewer has doubts about the analogy with earthquakes partly because of the lack of information on the location or spatio-temporal correlations of the events. We agree with the Reviewer that this information is crucial for the comparison. However, in the following we argue that we do have information on such correlations. To start with, strain bursts proceed in discrete events localized in distinct slip bands. This is clearly visible in Supplementary Video 1, where one may conclude by visual inspection that during a stress drop deformation indeed takes place in a single slip band. Since our experiments revealed that a stress drop is, in fact, a sequence of avalanches as measured by AE, we conclude that these AE events originate from the same slip band. We considered this finding as a proof for spatio-temporal correlations, and to be one supporting the analogy with earthquakes. This was the reason for conducting the edge detection described in the Methods section where we analysed a single event that confirmed the result of the visual inspection.

On the other hand, we agree with the Reviewer that this analysis should have been performed in a more general and convincing way. So, we extended the manuscript with a detailed analysis on the localization of the plastic deformation during stress drops. Please find details below, especially in our answer to comment #11.

Regarding the second concern about the similarity between fore-shock and after-shock rates, and its inconsistency with earthquakes, we would like to thank the Reviewer for these comments. In the revised manuscript this discrepancy is clearly discussed, in particular, when it comes to the analogy with earthquakes. See also our response to comment #4 below.

We especially thank the Reviewer for her/his numerous detailed and specific comments that we address below.

RC: *1. L25-27: this is not always the case, and depends on the material (see ref. 28).*

AR: Indeed, the experiments of the referred papers were conducted on ice single crystals with HCP structure. We now state explicitly the material in the specific sentence.

It was found that in bulk ice single crystals the recorded AE signal is burst-like and the energy associated with individual bursts follows a scale-free distribution[18, 19].

RC: *2. L65-66, and more generally about the evolution of stress and strain during an “avalanche”. As shown on Extended Data Fig 6, the characteristics of the loading device implies that an avalanche is characterized by both a stress drop ?? and a strain increment, which is classical for such type of experiment on micropillars. Consequently, the stress drop is maybe not the best measure of the strain energy released during the avalanche. An estimate of this strain energy should combine the evolution of both stress and strain. If the slope of the descending parts of the curves of Extended data Fig 6 is constant, and is simply a measure of the stiffness of the machine, then the released energy should scale as $\Delta\sigma^2$. In any case, on figure 2, it might be more judicious to discuss the scaling between AE energy and this strain energy, not the stress drop itself. On this figure 2, the authors report a scaling $E_{AE} \sim [(\Delta\sigma)]^{1.2}$, which would mean $E_{AE} \sim (\Delta\sigma^2)^{0.6}$. In other words, this would suggest that the AE energy scales with, but is not proportional to, the released strain energy. This differs from what is generally assumed to interpret AE data in the lab. This is reminiscent of the debate about seismic efficiency of earthquakes (e.g. [Kanamori and Brodsky, 2004 [20]]). This point is essential for a comparison of dislocation avalanches as observed from AE, or directly from stress-strain curves in μm - to sub μm -systems. It would therefore merit, in my opinion, a more thorough investigation from this novel dataset (maybe complemented with DDD simulations ?).*

AR: The issue raised by the Referee is indeed a central question both for earthquakes and also for acoustic emission measurements. In the original manuscript, due to space limitations, we did not enter into this discussion in detail, but we agree with the Referee, that an analysis on the energetics of the AE signals would make the paper even more interesting.

We start by noticing that there are three relevant energy quantities one may look at. The first one we call *injected energy* and is the work done by the compression device during an event:

$$E_{\text{inj}} = \bar{F}\Delta s \approx \frac{\sigma_0 + \sigma_1}{2} A\Delta s, \quad (5)$$

where \bar{F} is the average force exerted by the device, Δs is the displacement of the punch tip, σ_0 and σ_1 are the stresses before and after the event and A stands for the cross-section of the sample. This expression is equivalent to Eq. (3.14) in the referenced review article [20] where it is termed *potential energy*. If we assume a quick event with a stress drop of $\Delta\sigma = \sigma_0 - \sigma_1$, then

$$E_{\text{inj}} \approx \frac{A}{k}(\sigma_0 - \Delta\sigma/2)\Delta\sigma \propto (\sigma_0 - \Delta\sigma/2)\Delta\sigma, \quad (6)$$

where k is the stiffness of the device. This scaling is different from the one suggested by the Referee (that is, $E \propto (\Delta\sigma)^2$), which would hold if the stress drop magnitude was comparable with the applied stress itself.

This work E_{inj} is not necessarily equal to the dissipated energy, since the energy stored as elastic energy may change during an event. The second relevant energetic quantity is, therefore, the *dissipated energy* that actually equals the change of the elastic stored energy of the system (in this sense, the external work initially increases the elastic energy of the sample part of which then gets released due to plastic processes). Assuming a $v \propto \sigma$ linear relationship between the local stress and the drag acting on dislocations, the change in the

elastic energy can be written in the form:

$$E_{\text{dis}} \propto \sum_{i=1}^N \int_{t_0}^{t_1} v_i^2 l_i dt, \quad (7)$$

where t_0 and t_1 mark the beginning and the end of an event, respectively, and v_i and l_i are the velocity and length of the i -th dislocation or dislocation segment, respectively. This formula is the analogue of Eq. (3.12) of [20], that is termed *radiated energy* for earthquakes.

The dissipated energy may create, e.g., heat or elastic waves. With an acoustic transducer some part of the energy released in the form of elastic waves can be detected (denoted by E in the manuscript). Here we arrive at the issue of efficiency mentioned by the Referee. It is, of course, not known what portion of the dissipated energy gets converted into elastic waves and what fraction of it can be measured at the surface. Here we make the simplest and most straightforward assumption that this ratio does not depend on the energy of the event, so, the measured AE energies are representative of the released energy: $E \propto E_{\text{dis}}$. We note that our situation is somewhat simpler than earthquakes since here only one type of deformation is active contrary to earthquakes where rupture and thermally activated processes may also play a role.

Concerning the relationship between E_{inj} and E_{dis} the natural assumption is again a linear dependency in the average sense. This idea, however, can be tested by the DDD simulations as there we have direct access to microstructural data, i.e., dislocation positions and velocities. It is evident from Fig. 2 below that $E_{\text{dis}} \propto E_{\text{inj}}$ holds for events with large energies, but scaling with $(\Delta\sigma')^2$ is not linear and yields a larger scatter for large events. In addition, events with larger stresses (i.e., with brownish colour) dissipate more energy which is not accounted for if only $(\Delta\sigma')^2$ is considered as variable.

Figure 2: **a**, Relationship between the injected energy E_{inj} (that is, external work) and the dissipated elastic energy E_{dis} for individual events in 2D DDD simulations. **b**, Identical plot with squared stress drop size as variable. A clear deviation from the linear relationship is observed.

In the case of experiments only the injected energy E_{inj} and the detected acoustic energy E can be measured. In Fig. 3 we plot the two quantities against each other for individual events. As seen, in this case a linear relationship between E_{inj} and E is obtained that seem to match the assumptions mentioned above. It is noted, that in the original manuscript the stress drop was used as a variable because during experiments only a weak hardening is observed, so the stress σ does not vary much. The current re-analysis of the data, suggested by the Referee, clearly shows that the variations of the stress do play some role, and the consideration of this

effect leads to a better understanding of the energetics of dislocation avalanches. We also note, that scaling against $(\Delta\sigma)^2$, similarly to the DDD simulations mentioned above, is not conclusive (see Fig. 3(b)).

Figure 3: **a**, Relationship between the injected energy E_{inj} (that is, external work) and the measured AE energy E for individual events in microcompression experiments. **b**, Identical plot with squared stress drop size as variable. A clear deviation from the linear relationship is observed.

Several modifications were made in the resubmitted material. Firstly, the corresponding paragraph on experiments in the main text was modified as:

The facts that (i) stress drops $\Delta\sigma$ and AE energies E are detected in a correlated manner, (ii) both obey a scale-free distribution and (iii) the exponents are relatively close to each other suggest that there is a physical relation between them. To shed light on such a link, Fig. 2c provides a scatter plot of $\Delta\sigma$ and E corresponding to the individual stress drops (given that at least one AE event was detected during the stress drop) for $d = 32 \mu\text{m}$ pillars (for smaller pillar sizes see Extended Data Fig. 3). As said above, several AE events may be detected during a single drop. In such cases the energies of the corresponding AE events are added. As seen, there is a large scatter between $\Delta\sigma$ and E but, clearly, larger stress drops tend to emit signals with larger energies also expressed by the Pearson correlation found to be 0.5 ± 0.1 . If one, however, bins the data with respect to the stress drop size a clear power-law relation $E \propto \Delta\sigma^\eta$ is obtained between the two quantities with $\eta = 1.2 \pm 0.2$. This can be rationalized by assuming $P(E)dE = P(\Delta\sigma)d\Delta\sigma$ yielding the relation $\eta = (\tau_\sigma - 1)/(\tau_E - 1)$ which is fulfilled by the measured exponents within error margins. This means that although a one-to-one correspondence between E and $\Delta\sigma$ does not exist, there is a close to linear relationship in the average sense that allows one to obtain the distribution of the stress drops (that are proportional to the plastic strain induced by the local deformation event) from the statistics of the AE events.

The facts that (i) stress drops $\Delta\sigma$ and AE energies E are detected in a correlated manner, (ii) both obey a scale-free distribution and (iii) the exponents are relatively close to each other suggest that there is a physical relation between them. To shed light on such a link, Fig. 2c provides a scatter plot of the injected energy E_{inj} and the detected AE energy E corresponding to the individual stress drops (given that at least one AE event was detected during the stress drop) for $d = 32 \mu\text{m}$ pillars (for smaller pillar sizes see Extended Data Fig. 3). The injected energy refers to the work done by the compression device during a stress drop and is proportional with $E_{inj} \propto (\sigma - \Delta\sigma/2)\Delta\sigma$ with σ being the applied stress at the onset of the event (see ‘Energetic considerations’ in Methods for details and background discussion). As said above, several AE events may be detected during a single drop. In such cases the

energies of the corresponding AE events are added. As seen, there is a large scatter between $\Delta\sigma$ and E but, clearly, stress drops with larger injected energy E_{inj} tend to emit AE signals with larger energies also expressed by the Pearson correlation found to be 0.5 ± 0.1 . If one, however, bins the data with respect to the injected energy a clear close-to-linear dependence $E \propto E_{inj}$ is obtained between the two quantities. This means that although a one-to-one correspondence between E and E_{inj} does not exist, there is a linear relationship in the average sense that allows one to obtain the distribution of the local energy release (that is proportional to the plastic strain increment multiplied with the local stress during a local deformation event) from the statistics of the AE events.

Secondly, the corresponding paragraph on simulations in the main text was modified as:

It has been known that size distribution of dislocation avalanches exhibits a different exponent in simulations compared to the real samples [21, 22], yet, the temporal clustering of the simulated AE events shows very similar behaviour to experiments in terms of the correlation between **stress-jumps injected energies** and the AE energies (Fig. 4c), Omori law (Fig. 4d), and waiting time distribution (Fig. 4e).

Thirdly, Figs. 2c, 4c and Extended Data Fig. 3 were replaced according to our answer above. The corresponding captions were also adjusted. Finally, the discussion above is now included in the Methods as a new subsection ‘Energetic considerations’.

RC: 3. *L75-77 and Figures 1c to 1e: these key figures are too small, and so it is difficult for the reader to appreciate this discussion. I might suggest to separate these figures from figure 1a. It might be also useful to represent as well the AE activity vs the strain (not time); see comment #2 above.*

We agree with the Referee that Figures 1c-1e are crucial for the message of the paper. They would indeed deserve to be separated, they were grouped with panels 1a and 1b only because Nature Communications prefers the number of displayed items not to exceed 4 and they also belong together conceptually. In the revised manuscript we rearranged the panels of Fig. 1 to increase the size of panels 1c-1e.

Concerning the second suggestion of the Referee we agree that it would be useful to use strain as a variable instead of time. The problem is, that the sampling rate of the strain measurement (200 Hz) is significantly lower than that of the AE measurements (2.5 MHz). Since the AE events indeed take place on a shorter time-scale than the resolution of strain measurements (see Fig. 1e), one could only plot the AE signals on a stress-strain curve using some interpolation which, in our opinion, could be misleading in the representation of the measured data. We, therefore, kept time as a variable in Figs. 1c-1e, and refer the Referee to Extended Data Fig. 7 for the stress-strain curves of the measurements. We also include below the stress-strain curve of the same pillar investigated in Figs. 1c-1e.

RC: 4. *Figure 3, and aftershock analysis. The results reported on figure 3 are a clear signature of time correlations in dislocations dynamics. Such type of correlations were already characterized in another hexagonal material at bulk scale [Weiss and Miguel, 2004 [23]]. However, correlation is distinct from causality, i.e., in this context, from mechanical triggering. In case of earthquakes, the causal structure can be inferred from seismic data using a specific methodology (see [Marsan and Lengliné, 2008 [24]] or [Houdoux et al., 2021 [25]]). However, this requires an information about the location of the slip events, which is not available here. It is therefore difficult to discuss these results in terms of “aftershock triggering” stricto sensu. In addition, a comparison of figures 3a and 3c show a rather symmetric pattern, i.e. a “foreshock rate” close to the “aftershock rate”. This is fundamentally different from earthquakes. This might result from the ambiguity to define “mainshocks” from the present data. In conclusion, these*

Figure 4: Stress-strain curve of the micropillar investigated in Fig. 1c-1e in the main text. The inset plots a magnified view of a section of the main curve. As seen, stress drops are here seen as straight lines with a slope characteristic to the spring constant of the device (shown also by the grey parallel curves for clarity).

data reveal time correlations in dislocation dynamics, characterized by a power law decay, up to a time scale of the order of the duration of stress drops, i.e. “avalanches” defined from the stress-strain curves. Concerning this power law decay, another point should be scrutinized: owing to the loading apparatus (and its intrinsic stiffness) as well as the finite sample size, the triggering of an avalanche implies a stress drop during the avalanches (as observed here). Considering that, at the local level, the stress controls the unpinning of dislocations as well as their velocity, this decaying stress must have an impact on the rate of dislocation activity during such a stress drop. In other words, there is a coupling between dislocation avalanches and the dynamics of the global elastic system, and it would be useful to analyze the potential impact of this coupling on the time correlations reported above. More generally, these results reveal a two-levels structure of short bursts (measured from AE and correlated in time) embedded within larger instabilities leading to system-size stress drops. There is a clear link between large AE rates and these large instabilities, both AE bursts and large avalanches are power-law distributed, while AE bursts are correlated in time, but not the large instabilities. All of this is potentially of great interest for our understanding of dislocation dynamics in general, more particularly in μm - to sub- μm systems, and for the interpretation of micropillar compression tests. In my opinion, this would deserve a longer discussion in this manuscript.

AR: We sincerely thank the Referee for these suggestions that intend to help in deepening the discussion of the paper to make it even more interesting. Indeed, spatial correlations are crucial in the comparison with earthquakes. As the Referee says, an analysis analogous with [Marsan and Lengliné, 2008 [24]] is impossible since the exact location of the origin of the AE signals is not possible to determine in such a small volume and with a single transducer. However, we still believe that spatial correlations are present and that the analogy with earthquakes is not solely based on the similarities in the fundamental empirical laws. As detailed in the original manuscript, the cascade of correlated AE events comes from a single stress drop. These stress drops, on the other hand, typically correspond to the appearance of one slip band on the surface of the pillar. This statement was based on the visual inspection of Supplementary Video 1 and on the analysis of a single stress drop in Fig. 1b (the corresponding discussion can be found in the section ‘Edge detection’ of Methods). Since the source of AE signals can be assumed to be plastic deformation, these two observations clearly suggest that correlated AE events originate from a single slip band, in this sense they are spatially not random but strongly correlated. As said above, an even more detailed analysis on the correlations is impossible with the current experimental capabilities.

We, however, agree with the Referee that the second statement, that was based on visual inspection and the analysis of a single event, should be supported by somewhat stronger arguments. In addition, spatial correlations and their importance were not really discussed in the original manuscript.

Spurred by the Referee, we analysed the evolution of the micropillar surface to quantify the statement of ‘every stress drop corresponds to the appearance of a single slip band’. The details of the analysis are summarized below in our response to the Referee comment #9.

Regarding the asymmetry between the foreshock and aftershock rates we agree with the Referee, that although there is a difference of a factor of approximately 2 between the two rates, but this is significantly smaller than that seen for earthquakes. We also agree that the reason for this could be the ambiguity to define ‘mainshocks’ from the present data, as the measured AE energies may be the subject of larger error than the energies measured for earthquakes. This issue is now discussed in detail in the revised manuscript. We also included the missing reference [23] mentioned by the Referee:

As a further proof of equivalence, Fig. 3c plots the correspondent of the ‘inverse Omori law’ describing the power-law increase in the rate of foreshocks r_{fs} before a main shock [26]. (Extended Data Fig. 4 shows the corresponding figures for smaller pillars.) Previously, a similar analysis of AE on hexagonal ice at the bulk scale also showed more abundant triggering after large events, but the scale-free characteristics presented here were not possible to obtain, likely because of the relatively high level of noise [23]. It is also noted, that the difference between the foreshock and aftershock rates is significantly smaller compared to that of earthquakes. It is speculated, that the difference is caused by the ambiguity in the determination of the main shock due to the noise present in the AE energy measurement.

As for the last comment of the Referee regarding the physics of the subsequent triggering during an event cluster that is seen as a single stress drop we argue as follows. The Referee is right that during stress drops the driving force (that is, the applied stress itself) gets smaller that is expected to reduce the probability of subsequent triggering. In particular, the stress drop cannot be larger than the actual stress itself, so, there is definitely a hard barrier related to the stress. However, according to Fig. 2a the stress drops get smaller for larger systems that indicate that the stress may not be the limiting factor for avalanche propagation. As shown by the inset, the cut-off in force F_0 is independent of the specimen height $L = 3d$, which then also holds for the elongation increments $x_0 = F_0/k$, with k being the spring constant of the device. The effect of machine stiffness on the avalanche cut-off was investigated by Csikor *et al.* [27], they found that the cut-off in *strain* obeys scaling:

$$\varepsilon_0 \propto \frac{bE}{L(\Theta + \Gamma)}, \quad (8)$$

where Θ and Γ are the strain hardening coefficient and the machine stiffness, respectively. In our case the machine stiffness is $\Gamma \approx k/L$, since the elastic deformation of the pillar is negligible compared to that of the spring. Hence, the cut-off in force reads as

$$F_0 = kx_0 = kL\varepsilon_0 \propto bE \frac{k}{\Theta + k/L}. \quad (9)$$

The finding that F_0 does not depend on L suggests that Θ is significantly larger than k/L . But if we look at the approximate values we see that $k/L \approx 100$ MPa (for a 32 μm pillar) and the average slope of the stress-strain curves (see Fig. 4 in this response above) is around 50 MPa. To overcome this apparent contradiction we consider the origin of Θ in the scaling relation above. During a stress drop not only the stress decreases but also the material gets harder (strain hardening), and both processes act to cease the event. This is the reason

the sum of Θ and Γ appears above in Eq. (8). In Ref. [27] Θ was identified with the slope of the stress-strain curve. We believe, however, that this value is a local quantity and may differ from the global slope. During the compression of a pillar deformation proceeds in different shear bands. This can be envisaged as a weakest link process, that is, always the shear band with the lowest yield stress gets activated. After activation strain accumulates but the deformation stops and another shear band will get activated subsequently. This means that the yield stress of the activated shear band increases, that is, strain hardening takes place. This hardening coefficient of this local mechanism has nothing to do with the global coefficient, that also depends on the number of shear bands and the distribution of the yield stresses of the shear bands. So, we argue, that the Θ in Eq. (8) may be significantly larger than the global strain hardening coefficient. This could explain why F_0 is independent of the system size and suggests that the local hardening mechanism is dominant in the avalanche cut-off over the stress decrease due to the applied spring. Whether local hardening is due to dislocation accumulation or dislocation starvation through the surface is an open question that future TEM investigations are expected to answer.

The following paragraph was added to the manuscript on the behaviour of the cut-off and the role of stress relaxation during an event:

As seen in Fig. 2a the stress drop size distribution exhibits a cut-off, that is, correlated consecutive triggering during a strain burst is limited. The behaviour of the cut-off, thus, may shed light on the physics of the triggering mechanisms. As it was mentioned, the cut-off in stress σ_0 decreases with increasing pillar diameter d , but the cut-off in force $F_0 = d^2 \sigma_0$ is independent of d . According to Csikor *et al.* the physical origin of the cut-off is either elastic coupling with the compression device (during an event the applied stress drops that reduces the driving force of the event) or the strain hardening of the material (during an event plastic deformation makes the material harder so the driving force drops in a relative sense) [27]. From the comparison of the behaviour of the cut-off with the predictions of Csikor *et al.* we conclude that it is not the machine stiffness but rather the local strain hardening in the activated slip band is responsible for stopping the consecutive triggering taking place during a stress drop (see Methods for a detailed discussion).

In addition, the ‘Methods’ section was extended with a subsection that contains the extended discussion on the subject.

RC: *5. Figures 3e and 3f. These figures suggest that AE events are correlated in applied strain, not time. A similar problem was recently explored for the deformation of granular media [Houdoux et al., 2021 [25]]. This distinction is not just semantic; it might be important in terms of underlying physics (in others words, does time really matter here?).*

AR: We again thank the Referee for this very interesting comment and for drawing our attention to the paper of Houdoux *et al.* As we argued in the main text the waiting time distributions in Figs. 3e and 3f can be split into two parts: the power law regime of small waiting times with exponent 1.2 corresponds to AE signals originating from the same plastic event (and they are correlated) whereas the Poisson distribution of larger waiting times correspond to uncorrelated signals coming from different plastic events. As the platen velocity is decreased, the rate of plastic events (stress drops) decreases, too, which leads to the proportional increase of the waiting times. So, in this sense this is true that strain matters, but here we are talking about the uncorrelated, that is, independent events, so, this is not related to any kind of relaxation or triggering.

As far as correlated events are concerned where subsequent triggering is assumed to take place one should inspect the power-law part of the recurrence time distributions. Here, however, the curves overlap both before and after re-scaling because of the exponent being close to 1. So from the present data it seems not possible

to decide whether time or strain matters here. We included the following brief discussion on this matter in the revised manuscript at the end of Sec. ‘Aftershock and foreshock statistics’.

Recently, Houdoux *et al.* investigated the plastic response of granular systems that also showed remarkable analogy with earthquakes [28]. They concluded that in the triggering of subsequent events it is not the time but rather the strain that matters. In our case, however, because the exponent $1 - \gamma$ is rather close to one, one cannot decide whether time (Fig. 3e) or strain (Fig. 3f) matters in the triggering mechanism.

RC: 6. *Modelling: The DDD simulations are run using periodic boundary conditions. However, in μm to sub- μm pillars, finite size effects are likely playing an important role, changing the nature of dislocation sources, of nucleation mechanisms, etc. A comment on this would be welcome.*

AR: Indeed, in nanopillars it was found that exhaustion hardening as well as dislocation source truncation represent key physics in the plastic deformation that lead to size-effects. These single-dislocation properties are important at scales comparable or smaller than the average dislocation spacing, being around $\rho^{-0.5} \approx 0.1 \mu\text{m}$. In our experiments the micropillars are rather large ($8 - 32 \mu\text{m}$) compared to previous studies, and at this scale collective dislocation dynamics is expected to dominate plasticity and boundary effects can be neglected. This explains our choice for the PBCs that allowed us to concentrate on collective dislocation phenomena.

The following paragraph was added to Sec. ‘Discrete dislocation dynamics’ in the Methods:

With the application of the PBCs surface effects that may be important for small scale samples are neglected in the simulations. Indeed, in nanopillars it was found that exhaustion hardening [16] as well as dislocation source truncation [17] represent key physics in the plastic deformation that lead to size-effects. These single-dislocation properties are important at scales comparable or smaller than the average dislocation spacing, being around $\rho^{-0.5} \approx 0.1 \mu\text{m}$. In our experiments the micropillars are rather large ($8 - 32 \mu\text{m}$) compared to previous studies, and at this scale collective dislocation dynamics is expected to dominate plasticity and boundary effects can be neglected. This explains our choice for the PBCs that allowed us to concentrate on collective dislocation phenomena. It is also noted, that in experiments plastic deformation leads to the shape change of the sample and may cause some lattice rotation. These effects are not expected to have a significant role in the observed dynamics and are neglected in the simulations.

RC: 7. *L204-205: OK for HCP materials. This might be quite different for FCC or BCC materials, for which short-range mechanisms are expected to play a significant role.*

AR: We referred here to the current situation of Zn micropillars, but we agree with the Referee that in materials with different crystal structure this behaviour may differ. We clarified the referred sentence as:

We thus conclude, that the complex dynamic behaviour observed in the experiments reported in this paper is the result of the spatio-temporal correlations of the dislocations due to their long-range elastic interactions and the lack of short-range mechanisms, such as dislocation reactions.

For the discussion on differences between HCP and FCC/BCC materials see the following comment.

RC: 8. *L211-212: “and most likely crystalline metals/solids in general”. I disagree on this statement. FCC and BCC metals do not exhibit such scale-free dynamics in general [Weiss et al., 2021 [29]; Zhang et al., 2017 [14]]*

AR: We again agree with the Referee that this is a somewhat misleading formulation. We emphasize that we did mention in the original manuscript that multiple slip (and high temperature) leads to different critical behaviour. In the resubmitted material we removed the criticised misleading comment, explicitly stated the influence of crystal structure and made a reference to the paper mentioned by the Referee. The performed modifications are as follows:

The experiments and simulations reported here prove this long-standing hypothesis and reinforce that intermittency and scale-invariance characterizing plastic deformation of HCP single crystals (~~and, most likely, crystalline metals/solids in general~~) are related to the self-organized critical (SOC) behaviour of dislocations. In addition, we showed that plastic events, similarly to earthquakes, do not only exhibit spatial but also temporal clustering with long-range correlations, however, the involved length and timescales are profoundly different, as summarized in Extended Data Table 2. This phenomenon also raises analogy with many other physical systems exhibiting crackling noise [12]. It is known, however, that SOC behaviour is not ubiquitous in crystal plasticity, for instance, it is suppressed in materials with FCC and BCC crystal structure and under multiple slip conditions likely because of short-range interactions related to dislocation reactions and also at high temperatures [13, 14, 15].

RC: **9. L214: spatial correlations were not analyzed here.**

AR: Please see our response below for comment #11.

RC: **10. L302: Could the authors give the corresponding applied strain-rate ?**

AR: The platen velocity during the compression tests was (apart from the analysis focusing on the waiting time statistics) constant ($v_p = 10$ nm/s), therefore the strain rates for the three different sized micropillars were between $\dot{\epsilon} = 1 \times 10^{-4} - 4 \times 10^{-4}$ 1/s.

RC: **11. Section on “Edge detection”. This section describes a nice example showing the activation of a single slip band associated to a single stress drop (see also Fig 1). It is however not clear whether the authors performed such image analysis at a regular periodicity ? or just for few examples of stress drops.**

AR: As we mentioned above in our response to Referee comment #4, the edge detection was performed for one stress drop, and we made our conclusion that stress drops correspond to the appearance of a single slip band based on the visual inspection of Supplementary Video 1.

We agree, however, that this conjecture is crucial for the comparison with earthquakes (as spatial correlations are inherent properties of earthquake clusters), therefore, a quantitative analysis is required that supports this finding. We have now, therefore, analysed the shape change of the micropillars for every plastic event during the course of a compression experiment using the SEM images. The analysis is based on the edge detection summarized in the Supplementary Material. The aim is to give an estimation on how much the plastic deformation occurring between two consecutive SEM images is localised, that is, taking place in a single slip band.

To this end, we started from the raw edge profile $x_{\text{raw}}(z, t)$ obtained from the edge detection. This curve contains outliers due to the noise present in the backscattered electron images of the SEM. To reduce this noise we applied a median based smoothing algorithm: If the distance of a point from the median of its upper and lower neighbours (7 – 7 pixels) was greater than $2 \mu\text{m}$, it was replaced by the median. Otherwise the original value was kept. The resulting curve is denoted by $x(z, t)$. This method efficiently reduced the noise as seen in the updated Extended Data Fig. 8 (curves with colours from yellow to blue). Next, the difference between consecutive images was computed as $\Delta x_{\text{raw}}(z, t) = x(z, t + \Delta t) - x(z, t)$, where Δt is the time

needed to record an SEM image (~ 0.25 s in our case). An exemplary of such a differential curve is seen in Extended Data Fig. 8 (pink curve on the right-hand-side).

This curve was still rather noisy to determine how localized the deformation was. We, therefore, calculated the moving average with a window size of 15 pixels and then applied a moving median smoothing with a window size of 61 pixels. Finally, we made the curves monotonous, since slip in the opposite direction was not observed in the experiments. Three representative exemplary so-obtained $\Delta x(z, t)$ curves can be seen below in the bottom row of Fig. 5.

The obtained profiles usually exhibit a single slip band, but sometimes more than one step in the profile is seen. To quantify to what extent is the deformation localized we use the method of Ref. [30]. Namely, we first note, that $\Delta x(z, t)$ is defined on an equidistant grid of the individual pixels of the SEM image. Let the discretized profile be denoted as Δx_i (for simplicity we omit the reference to time t). The local strain increment is then $\Delta \varepsilon_i^{\text{pl}} = \Delta x_{i+1} - \Delta x_i$. We now select an arbitrary point with index k along the height of the pillar and consider the typical distance of the plastic strain increments from this point as:

$$d_k = \frac{\sum_i \Delta \varepsilon_i^{\text{pl}} |k - i| \Delta z}{\sum_i \Delta \varepsilon_i^{\text{pl}}}, \quad (10)$$

where Δz is the pixel size of the SEM image. Then the minimum $d_{\min} = \min_k d_k$ is determined. For a homogeneous distribution of the plastic strain (i.e., $\Delta x(z)$ is a linear function) d_{\min} equals $L/4$, where L is the height of the micropillar. On the other hand, for a fully localized strain distribution (i.e., $\Delta x(z)$ is a step function) $d_{\min} = 0$ is obtained. The localization parameter is, therefore, defined as

$$\eta = 1 - \frac{4}{L} d_{\min}. \quad (11)$$

Consequently, $\eta = 0$ signals a homogeneous deformation, whereas $\eta = 1$ is characteristic of deformation fully localized in a single slip band.

Figure 5 summarizes the analysis performed on those consecutive images, where the event size defined as $\Delta x(L) - \Delta x(0)$ (that is, the displacement between the top and bottom of the pillar) was larger than $0.02 \mu\text{m}$. As seen the localization η is typically between 0.5 and 1 and its average is $\langle \eta \rangle = 0.74$. This high value of η clearly shows that plastic strain increments are quite localized. The fact that the values are smaller than 1 are likely due to the numerical noise present in the edge detection and that between two consecutive SEM images (that takes around 0.25 s) more than one events can take place.

So, based on this analysis we conclude that the plastic strain increments are highly localized, typically concentrating in a single slip band. Since the AE events are observed during the accumulation of plastic strain, it is natural to assume that a cascade of events correlated in time originate from the same (typically one, sometimes few) slip band. This means these events are not only correlated in time, but also in space. More details on the spatial correlations are not possible to obtain with the present experimental methods due to the small volume of the specimen and the time resolution of the SEM.

Based on this discussion the following paragraph was added to the main text of the manuscript:

These results unveil an interesting two-level structure of plastic activity: accumulation of plastic strain is characterized by the intermittent appearance of uncorrelated slip bands. These strain bursts induce stress drops due to the stiffness of the compression device. But AE measurements reveal that these strain bursts themselves are characterized by a sequence of local events with complex spatiotemporal dynamics. Firstly, Omori law and the waiting time distributions report about scale-free temporal

Figure 5: Top row: Localization parameter η as a function of time t and event size $\Delta x(L) - \Delta x(0)$. The color scale refers to the event size in the left panel. The right panel plots the distribution of the localization parameter η . Bottom row: Three exemplary $\Delta x_{\text{raw}}(z, t)$ (light gray) and the corresponding $\Delta x(z, t)$ (blue) profiles obtained using the method described in the text. The datapoints corresponding to the curves are circled with the same colour as the frame of the figures.

correlations. Secondly, since strain increments during stress drops are localized in distinct slip bands (see Supplementary video 1) and plastic activity was found to be responsible for AE, one can conclude that the correlated AE events originate from a single slip band, that is, they are not only temporally but also spatially correlated. To quantify this observation the strain evolution between subsequent SEM images during the *in situ* compression was analyzed in the Methods (see section ‘Strain localization’) and we concluded that deformation during a single stress drop is indeed highly localized and is typically concentrated in one or sometimes few individual slip bands.

In addition the detailed discussion of this response was added to Methods as a separate subsection ‘Strain localization’. Figure 5 was also added to methods as Extended Data Fig. 10. Finally a new Supplementary Video 4 was also added that demonstrates how the edge detection algorithm works.

RC: *12. Extended figure 13: it would nice to show the evolution of the stress as well here.*

AR: The image, now called Extended Data Fig. 15, has been modified in the revised material and the evolution of the stress is now also shown along the energy dissipation rate.

RC: *Refs.:*

Houdoux, D., A. Amon, D. Marsan, J. Weiss, and J. Crassous (2021), *Micro-slips in an experimental granular shear band replicate the spatiotemporal characteristics of natural earthquakes*, *Communications Earth & Environment*, 2(1), 1-11.

Kanamori, H., and E. E. Brodsky (2004), *The physics of earthquakes*, *Reports on Progress in Physics*, 67(8), 1429.

Marsan, D., and O. Lengliné (2008), *Extending earthquakes' reach through cascading*, *Science*, 319, 1076-1079.

Weiss, J., and M.-C. Miguel (2004), *Dislocation avalanche correlations*, *Mat. Sci. Eng. A*, 387-389(292-296).

Weiss, J., P. Zhang, O. U. Salman, G. Liu, and L. Truskinovsky (2021), *Fluctuations in crystalline plasticity*, *Comptes Rendus. Physique*, 22(S3), 1-37.

Zhang, P., O. G. Salman, J. Y. Zhang, G. Liu, J. Weiss, L. Truskinovsky, and J. Sun (2017), *Taming intermittent plasticity at small length scales*, *Acta Materiala*, 128, 351-364.

AR: We thank the Reviewer for raising our attention to these papers. They are now cited at the corresponding parts of the resubmitted paper.

References

- [1] Groma, I. X-ray line broadening due to an inhomogeneous dislocation distribution. *Physical Review B* **57**, 7535 (1998).
- [2] Borbély, A. & Groma, I. Variance method for the evaluation of particle size and dislocation density from x-ray bragg peaks. *Applied Physics Letters* **79**, 1772–1774 (2001).
- [3] Groma, I. & Borbély, A. X-ray peak broadening due to inhomogeneous dislocation distributions. In *Diffraction Analysis of the Microstructure of Materials*, 287–307 (Springer, 2004).
- [4] Teodosiu, C. *Elastic models of crystal defects* (Springer Science & Business Media, 2013).
- [5] Dragomir, I. C. & Ungár, T. Contrast factors of dislocations in the hexagonal crystal system. *Journal of Applied Crystallography* **35**, 556–564 (2002).
- [6] Borbély, A., Dragomir-Cernatescu, J., Ribárik, G. & Ungár, T. Computer program ANIZC for the calculation of diffraction contrast factors of dislocations in elastically anisotropic cubic, hexagonal and trigonal crystals. *Journal of Applied Crystallography* **36**, 160–162 (2003).
- [7] Britton, T. B. & Wilkinson, A. J. High resolution electron backscatter diffraction measurements of elastic strain variations in the presence of larger lattice rotations. *Ultramicroscopy* **114**, 82–95 (2012).
- [8] ISO 12716:2001(E): Non-destructive testing – Acoustic emission inspection – Vocabulary. Standard, International Organization for Standardization, Geneva, Switzerland (2001).
- [9] Grosse, C. & Ohtsu, M. (eds.) *Acoustic Emission Testing* (Springer Berlin Heidelberg, Berlin, Heidelberg, 2008).
- [10] The Japanese Society for Non-Destructive Inspection. *Practical Acoustic Emission Testing* (Springer Japan, Tokyo, 2016).
- [11] Hull, D. & Bacon, D. J. *Introduction to dislocations* (Butterworth-Heinemann, 2001).

- [12] Sethna, J. P., Dahmen, K. A. & Myers, C. R. Crackling noise. *Nature* **410**, 242–250 (2001).
- [13] Weiss, J. *et al.* From mild to wild fluctuations in crystal plasticity. *Phys. Rev. Lett.* **114**, 105504 (2015).
- [14] Zhang, P. *et al.* Taming intermittent plasticity at small length scales. *Acta Materialia* **128**, 351–364 (2017).
- [15] Alcalá, J. *et al.* Statistics of dislocation avalanches in FCC and BCC metals: dislocation mechanisms and mean swept distances across microsample sizes and temperatures. *Sci. Rep.* **10**, 1–14 (2020).
- [16] Shan, Z., Mishra, R. K., Asif, S. S., Warren, O. L. & Minor, A. M. Mechanical annealing and source-limited deformation in submicrometre-diameter ni crystals. *Nature materials* **7**, 115–119 (2008).
- [17] Tang, H., Schwarz, K. & Espinosa, H. Dislocation-source shutdown and the plastic behavior of single-crystal micropillars. *Physical review letters* **100**, 185503 (2008).
- [18] Miguel, M.-C., Vespignani, A., Zapperi, S., Weiss, J. & Grasso, J.-R. Intermittent dislocation flow in viscoplastic deformation. *Nature* **410**, 667–671 (2001).
- [19] Weiss, J. & Marsan, D. Three-dimensional mapping of dislocation avalanches: clustering and space/time coupling. *Science* **299**, 89–92 (2003).
- [20] Kanamori, H. & Brodsky, E. E. The physics of earthquakes. *Reports on Progress in Physics* **68**, 1429 (2004).
- [21] Ispánovity, P. D. *et al.* Avalanches in 2D dislocation systems: Plastic yielding is not depinning. *Phys. Rev. Lett.* **112**, 235501 (2014).
- [22] Lehtinen, A., Costantini, G., Alava, M. J., Zapperi, S. & Laurson, L. Glassy features of crystal plasticity. *Phys. Rev. B* **94**, 064101 (2016).
- [23] Weiss, J. & Miguel, M.-C. Dislocation avalanche correlations. *Mat. Sci. Eng. A* **387-389**, 292–296 (2004).
- [24] Marsan, D. & Lengliné, O. Extending earthquakes’ reach through cascading. *Science* **319**, 1076–1079 (2008).
- [25] Houdoux, D., Amon, A., Marsan, D., Weiss, J. & Crassous, J. Micro-slips in an experimental granular shear band replicate the spatiotemporal characteristics of natural earthquakes. *Communications Earth Environment* **2**, 1–11 (2021).
- [26] Jones, L. M. & Molnar, P. Some characteristics of foreshocks and their possible relationship to earthquake prediction and premonitory slip on faults. *J. Geophys. Res.-Sol. Ea.* **84**, 3596–3608 (1979).
- [27] Csikor, F. F., Motz, C., Weygand, D., Zaiser, M. & Zapperi, S. Dislocation avalanches, strain bursts, and the problem of plastic forming at the micrometer scale. *Science* **318**, 251–254 (2007).
- [28] Houdoux, D., Amon, A., Marsan, D., Weiss, J. & Crassous, J. Micro-slips in an experimental granular shear band replicate the spatiotemporal characteristics of natural earthquakes. *Communications Earth & Environment* **2**, 1–11 (2021).
- [29] Weiss, J., Zhang, P., Salman, O. U., Liu, G. & Truskinovsky, L. Fluctuations in crystalline plasticity. *Comptes Rendus Physique* **22**, 1–37 (2021).

- [30] Tüzes, D., Ispánovity, P. D. & Zaiser, M. Disorder is good for you: the influence of local disorder on strain localization and ductility of strain softening materials. *International Journal of Fracture* **205**, 139–150 (2017). URL <https://doi.org/10.1007/s10704-017-0187-1>.

Reviewers' Comments:

Reviewer #1:

Remarks to the Author:

The authors presented a revised version of their manuscript. The concerns I had in the first version have been addressed. Thus, I feel that the manuscript is now complete and congratulate the authors to the impressive work.

Reviewer #2:

Remarks to the Author:

The acoustic emission AE given by dislocation avalanches resulted in strain burst in Zn crystal is analogous to earthquakes. It is a nice creativity topic to analogy AE in micron scale pillar to earthquakes in macro scale. The authors have answered my review's questions and provide modification in the revised manuscript. I agree with their responses.

Reviewer #3:

Remarks to the Author:

The authors carefully addressed the referees questions and comments. To do so, they performed very interesting additional analyses that enrich the manuscript. I particularly appreciated the discussion about the scaling between the injected energy and the AE energy. It is also quite clear now that AE avalanches embedded within a single stress drop correspond to microslips along a single slip band. I am less convinced about the discussion on the role of strain-hardening on the avalanche cut-off, especially in such HCP material where classical hardening is unexpected, but this is probably a minor point. Consequently, I congratulate the authors for this very nice piece of work, which certainly improves our understanding of plasticity at the micron-scale, and deserves publication in Nature Communications.

On the other hand, I am still not fully convinced by the strict analogy between these slip events and earthquakes stressed by the authors in the title and the abstract. Also this would have been, maybe, slightly less "appealing", I would have instead insist already in the abstract about this 2-levels structure of microslips (revealed from AE) within individual stress drops (identified as single avalanches in classical micropillar analyses), which is in my opinion a more important point. However, I leave the editors to decide about this final presentation.

Authors' Response to Reviews of

Dislocation Avalanches: Earthquakes on the Micron Scale

Péter Dusán Ispánovity, Dávid Ugi, Gábor Péterffy, Michal Knappek, Szilvia Kalácska, Dániel Tüzes, Zoltán Dankházi, Kristián Máthis, František Chmelfík, István Groma
Nature Communications, NCOMMS-21-22888A

RC: Reviewers' Comment, AR: Authors' Response, Manuscript Text

WE THANK THE REVIEWERS FOR THEIR VALUABLE THOUGHTS AND COMMENTS ON THIS WORK, AND FOR RECOMMENDING THE MANUSCRIPT FOR PUBLICATION. OUR RESPONSE IS PROVIDED BELOW. CHANGES TO THE ORIGINAL MANUSCRIPT HAVE BEEN HIGHLIGHTED (~~DELETED PARTS~~ , ADDED PARTS) IN THIS DOCUMENT AND TRACED BY "LATEXDIFF" IN THE REVISED FILE.

Reviewer #1

RC: *The authors presented a revised version of their manuscript. The concerns I had in the first version have been addressed. Thus, I feel that the manuscript is now complete and congratulate the authors to the impressive work.*

AR: We thank the Reviewer for her/his overall evaluation, for recommending our work for publication and for her/his contribution in improving the paper.

Reviewer #2

RC: *The acoustic emission AE given by dislocation avalanches resulted in strain burst in Zn crystal is analogous to earthquakes. It is a nice creativity topic to analogy AE in micron scale pillar to earthquakes in macro scale. The authors have answered my review's questions and provide modification in the revised manuscript. I agree with their responses.*

AR: We thank the Reviewer for her/his overall evaluation, for recommending our work for publication and for her/his contribution in improving the paper.

Reviewer #3

RC: *The authors carefully addressed the referees questions and comments. To do so, they performed very interesting additional analyses that enrich the manuscript. I particularly appreciated the discussion about the scaling between the injected energy and the AE energy. It is also quite clear now that AE avalanches embedded within a single stress drop correspond to microslips along a single slip band. I am less convinced about the discussion on the role of strain-hardening on the avalanche cut-off, especially in such HCP material where classical hardening is unexpected, but this is probably a minor point. Consequently, I congratulate the authors for this very nice piece of work, which certainly improves our understanding of plasticity at the micron-scale, and deserves publication in Nature Communications. On the other hand, I*

am still not fully convinced by the strict analogy between these slip events and earthquakes stressed by the authors in the title and the abstract. Also this would have been, maybe, slightly less “appealing”, I would have instead insist already in the abstract about this 2-levels structure of microslips (revealed from AE) within individual stress drops (identified as single avalanches in classical micropillar analyses), which is in my opinion a more important point. However, I leave the editors to decide about this final presentation.

AR: We thank the Reviewer for her/his overall evaluation, for recommending our work for publication and for her/his contribution in improving the paper. We agree with the Reviewer to emphasize the two-level structure already in the abstract, which is indeed an important point. To follow this suggestion the abstract has been modified, see below. Note, that due to formatting requirements the abstract has also been slightly shortened, references have been removed and some phrases have been corrected.

~~Metals usually deform irreversibly as a result of the motion of dislocations that are line-like defects in the crystal lattice. Compression experiments of micron-scale specimens as well as acoustic emission (AE) measurements performed on bulk samples revealed that the motion of dislocations resembles a stick-slip process. As a result, deformation proceeds in a series of unpredictable local strain bursts with a scale-free size distribution. Here we use a unique, highly sensitive experimental set-up, which allows us to detect the weak AE waves of dislocation slip during the compression of micron-sized Zn pillars. This opens up new vistas for studying the stop-and-go dislocation motion in detail and understanding the physical origin of AE events. Profound correlation is observed between the energies associated with deformation events and with the emitted AE signals that, as we conclude, are induced by the collective dissipative motion of dislocations. We also show by statistical analyses of the acoustic event sequences that, despite of the fundamental differences in the deformation mechanism and the huge gap in the involved length and timescales, that in the studied material dislocation avalanches and earthquakes are essentially alike. Our experimental and computer simulation results not only unveil the complex spatiotemporal structure of strain bursts but also exhibit technological importance by unraveling the missing relationship between the properties of acoustic signals and the corresponding local deformation events.~~

Compression experiments on micron-scale specimens and acoustic emission (AE) measurements on bulk samples revealed that the dislocation motion resembles a stick-slip process – a series of unpredictable local strain bursts with a scale-free size distribution. Here we present a unique experimental set-up, which detects weak AE waves of dislocation slip during the compression of Zn micropillars. Profound correlation is observed between the energies of deformation events and the emitted AE signals that, as we conclude, are induced by the collective dissipative motion of dislocations. The AE data also reveal a surprising two-level structure of plastic events, which otherwise appear as a single stress drop. Hence, our experiments and simulations unravel the missing relationship between the properties of acoustic signals and the corresponding local deformation events. We further show by statistical analyses that despite fundamental differences in deformation mechanism and involved length- and time-scales, dislocation avalanches and earthquakes are essentially alike.